# Rotary Position Encodings for Graph-Structured Data

## Abstract

We introduce WIRE: Wavelet-Induced Rotary Encodings. WIRE extends Rotary Position Encodings (RoPE), a popular algorithm in LLMs and ViTs, to graph-structured data. We demonstrate that WIRE is more general than RoPE, recovering the latter in the special case of grid graphs. WIRE also enjoys a host of desirable theoretical properties, including equivariance under node ordering permutation, compatibility with linear attention, and (under select assumptions) asymptotic dependence on graph resistive distance. We test WIRE on a range of synthetic and real-world tasks, including identifying monochromatic subgraphs, semantic segmentation of point clouds, and more standard graph benchmarks. We find it to be effective in settings where the underlying graph structure is important.

## 1 Introduction

Position encodings incorporate information about the respective locations of tokens into the transformer attention mechanism (Vaswani et al., 2017). This is important because the meaning of a sequence of words or image patches in general depends upon how they are ordered. Likewise, the meaning of a graph depends upon how its constituent nodes are connected. Position encodings capture these spatial and topological relationships, enabling the network to learn expressive functions that generalise well to unseen data.

**APEs and RPEs**. Early transformers relied on absolute position encodings (APEs), which add or concatenate fixed or learned embeddings to each token (Kiyono et al., 2021; Liu et al., 2020; Wang et al., 2020). Whilst simple, these generally perform worse than relative position encodings (RPEs), which instead modulate attention logits for each query-key pair by a bias, taking $\boldsymbol{q}_i^\top \boldsymbol{k}_j \to \boldsymbol{q}_i^\top \boldsymbol{k}_j + b_{ij}$ (Li et al., 2023; Raffel et al., 2020; Shaw et al., 2018). The bias $b_{ij}$ depends on the tokens' respective positions, e.g. sequence separation in text or shortest path distance between graph nodes. Recent years have witnessed RPEs in turn be superseded by rotary position encodings (RoPE) (Su et al., 2024). RoPE decomposes tokens into 2-dimensional blocks and rotates them by position-dependent angles. RoPE's strong empirical performance and modest computational footprint have fuelled its growing popularity in LLMs and ViTs (Dubey et al., 2024; Gemma Team et al., 2024; Heo et al., 2024). Moreover, it enjoys the convenient property that (as with APEs) it directly modifies tokens, rather than the logits of query-key pairs. This makes RoPE compatible with linear attention and KV-caching, improving scalability with respect to the number of tokens.

**Position encodings for graphs**. Without a simple single 'coordinate system', position encodings for *graphs* — sets of nodes connected by edges — are more complicated. One choice is to use the spectrum of the graph Laplacian to build APEs (Dwivedi and Bresson, 2020; Kreuzer et al., 2021). In the special case of grid graphs, this closely resembles the sinusoidal APEs applied to text and images. Alternatively, one can compute some structural property like the shortest path distance or effective resistance for each pair of graph nodes, and use these quantities as RPE biases (Ying et al., 2021; Zhang et al., 2023). In this paper, we show how RoPE can be extended to graphs, providing a competitive and scalable alternative. Our algorithm mitigates some of the shortcomings of APEs and bias-based RPEs, encoding (approximate) invariances whilst preserving compatability with linear attention.

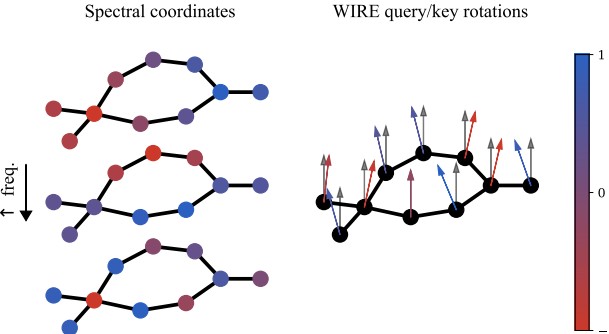

Spectral coordinates          WIRE query/key rotations

Figure 1: **WIRE schematic.** WIRE constructs spectral coordinates for each node, e.g. by computing the first few eigenvectors of the graph Laplacian. Low frequencies vary slowly across the graph; higher frequencies oscillate sharply between adjacent nodes. The spectral coordinates are projected down to obtain rotation angles for every query and key, applied in a RoPE-style position encoding (Su et al., 2024). WIRE enjoys desirable theoretical properties (Section 3.1) and is compatible with linear attention (Katharopoulos et al., 2020).

**Key contributions**. 1) We introduce **WIRE** (Wavelet-Induced Rotary Encodings), a new RoPE-style position encoding for graph-structured data. Figure 1 gives a schematic. 2) We show that WIRE is more general than RoPE, and that it can stochastically downweight attention scores based on graph effective resistance. 3) We demonstrate that WIRE is competitive in synthetic graph tasks, experiments with point clouds, and graph benchmarks.

## 2 PRELIMINARIES

Consider an undirected graph $\mathcal{G}(\mathcal{N}, \mathcal{E})$, where $\mathcal{N} := \{v_1, ..., v_N\}$ is a set of $N$ nodes and $\mathcal{E}$ is a set of edges. $(v_i, v_j) \in \mathcal{E}$ if and only if there exists an edge between $v_i$ and $v_j$ in $\mathcal{G}$. The number of nodes $N$ is equal to the number of tokens processed using a transformer. Let $\{x_i\}_{i=1}^N \subset \mathbb{R}^d$ denote this set of $d$-dimensional tokens. $d$ is assumed to be even.

**Attention**. The $i$th query, key and value vectors are given by $q_i = \mathbf{W}_q x_i$, $k_i = \mathbf{W}_k x_i$ and $v_i = \mathbf{W}_v x_i$ respectively, with $\mathbf{W}_q, \mathbf{W}_k, \mathbf{W}_v \in \mathbb{R}^{d \times d}$ learned projection matrices. For simplicity of notation we assume the single-head setting, with the understanding that all arguments are trivially generalised to multi-head attention. The *attention mechanism*, one of the fundamental computational units of the transformer, is written:

$$x_i \to \frac{\sum_j \mathrm{sim}(q_i, k_j) v_j}{\sum_{j'} \mathrm{sim}(q_i, k_{j'})}. \tag{1}$$

Here, $\mathrm{sim}(\cdot, \cdot) : \mathbb{R}^d \times \mathbb{R}^d \to \mathbb{R}$ is a 'similarity' function that assigns a score to each query-key pair. Standard softmax attention uses $\mathrm{sim}(q_i, k_j) = \exp(q_i^\top k_j)$, whereas linear attention takes $\mathrm{sim}(q_i, k_j) = q_i^\top k_j$ (Katharopoulos et al., 2020). The former generally works better, but the latter enables one to write a low-rank decomposition of the attention matrix, unlocking $\mathcal{O}(N)$ scaling. Concretely, with a slight abuse of notation, with linear attention one can take $x_i \to q_i^\top \left(\sum_j k_j v_j\right) / q_i^\top \left(\sum_{j'} k_{j'}\right)$. The commutativity of matrix-matrix multiplication obviates instantiating the attention matrix $[\mathrm{sim}(q_i, k_j)]_{i,j=1}^N \in \mathbb{R}^{N \times N}$ in memory. In the same spirit, one can define (random) feature maps $\varphi(\cdot) : \mathbb{R}^d \to \mathbb{R}^m$ and take $\mathrm{sim}(q_i, k_j) = \varphi(q_i)^\top \varphi(k_j)$, again unlocking $\mathcal{O}(N)$ scaling (Choromanski et al., 2020). Common choices for $\varphi(\cdot)$ include ReLU activations and random Laplace features (Yang et al., 2014).

**Rotary position encodings**. Suppose that each token is equipped with a $m$-dimensional coordinate $r_i \in \mathbb{R}^m$, with $m = 1$ for sequences, $m = 2$ for images and $m = 3$ for videos and point clouds. Given a (projected) token $z_i \in \{q_i, k_i\}$, RoPE takes $z_i \to \mathrm{RoPE}(r_i) z_i$, where:

$$\text{RoPE}(\boldsymbol{r}_i)\boldsymbol{z}_i := \bigoplus_{n=1}^{d/2} \boldsymbol{\rho}(\theta_n)[\boldsymbol{z}_i]_{2n-2:2n-1}, \quad \boldsymbol{\rho}(\theta) := \begin{pmatrix} \cos(\theta) & -\sin(\theta) \\ \sin(\theta) & \cos(\theta) \end{pmatrix}, \quad \theta_n := \boldsymbol{\omega}_n^\top \boldsymbol{r}_i. \quad (2)$$

Here, $\bigoplus$ denotes the direct product, so each $2 \times 2$ matrix $\boldsymbol{\rho}(\theta_n)$ rotates a 2-element section of the query or key. Meanwhile, $\{\boldsymbol{\omega}_n\}_{n=1}^{d/2} \subset \mathbb{R}^m$ are learnable or fixed frequencies.[1] Using the basic properties of 2D rotations, it is straightforward to see that

$$\text{RoPE}(\boldsymbol{r}_i)^\top \text{RoPE}(\boldsymbol{r}_j) = \text{RoPE}(\boldsymbol{r}_j - \boldsymbol{r}_i), \quad (3)$$

whereupon the joint transformation of queries and keys takes $\boldsymbol{q}_i^\top \boldsymbol{k}_j \to \boldsymbol{q}_i^\top \text{RoPE}(\boldsymbol{r}_j - \boldsymbol{r}_i)\boldsymbol{k}_j$. Clearly, RoPE is translationally invariant,[2] an inductive bias that helps it generalise to new sequence lengths and makes it effective in 3D robotics applications (Schenck et al., 2025).

**Transformers for graphs**. Whilst Graph Neural Networks (GNNs) have traditionally performed best for graph-structured data, recent years have witnessed growing interest in transformers (Müller et al., 2023; Veličković et al., 2017; Ying et al., 2021). A key algorithmic challenge is to design effective position encodings that capture important structural information about $\mathcal{G}$. To this end, researchers often consider graph spectra (Chung, 1997).

**Graph spectra**. Let $\mathbf{A} := \left[ \mathbb{I}\big((v_i, v_j) \in \mathcal{E}\big) \right]_{i,j=1}^N \in \{0, 1\}^{N \times N}$ denote the graph *adjacency matrix*, whose $(i, j)$ entry is equal to 1 if the corresponding edge is present in the graph and 0 otherwise. Let $\mathbf{D} := \text{diag}\big(\sum_j \mathbf{A}_{ij}\big)$ denote the diagonal degree matrix. The *graph Laplacian* is given by $\mathbf{L} := \mathbf{D} - \mathbf{A} \in \mathbb{R}^{N \times N}$. Since it is symmetric, we can write

$$\mathbf{L} = \mathbf{U}\boldsymbol{\Lambda}\mathbf{U}^\top, \quad \boldsymbol{\Lambda} = \text{diag}(\lambda_0, ..., \lambda_{N-1}), \quad (4)$$

with $\lambda_0 \leq \lambda_1 \leq ... \leq \lambda_{N-1}$. Here, $\mathbf{U} := [\boldsymbol{u}_0, \boldsymbol{u}_1, ..., \boldsymbol{u}_{N-1}]^\top$ is orthonormal, with each each eigenvector (column) $\boldsymbol{u}_i \in \mathbb{R}^N$ oscillating across the graph at frequency $\lambda_i$. The spectrum of $\mathbf{L}$ (or its normalised variant $\mathbf{D}^{-1/2}\mathbf{L}\mathbf{D}^{-1/2}$) captures the structure of $\mathcal{G}$. $\mathbf{U}$ and $\boldsymbol{\Lambda}$ are often used to construct graph transformer APEs. Here, we will use them within RoPE.

**Remainder of the manuscript**. In Section 3 we introduce Wavelet-Induced Rotary Encodings (WIRE), generalising RoPE to graphs. We show that WIRE enjoys a host of attractive theoretical properties. In Section 4, we demonstrate that WIRE performs competitively in learning tasks with a strong structural component.

## 3 WIRE: WAVELET-INDUCED ROTARY ENCODINGS

We begin by defining WIRE.

---

**Alg. 1. Wavelet-Induced Rotary Encodings (WIRE).** 🧵

1. Compute the lowest $m \leq N$ eigenvectors and eigenvalues $\{\boldsymbol{u}_k, \lambda_k\}_{k=0}^{m-1}$ of the graph Laplacian $\mathbf{L}$, either exactly or with approximate iterative methods.
2. Define *spectral features* for each graph node, e.g. $\boldsymbol{r}_i = [\boldsymbol{u}_k[i]]_{k=0}^{m-1} \in \mathbb{R}^m$ or similar.
3. Apply rotary position encodings using these spectral features, taking $\boldsymbol{z}_i \to \text{RoPE}(\boldsymbol{r}_i)\boldsymbol{z}_i$ for queries and keys $\boldsymbol{z}_i \in \{\boldsymbol{q}_i, \boldsymbol{k}_i\}$.

---

**Efficiency of WIRE**. Once equipped with spectral coordinates, WIRE becomes extremely efficient to compute. This is because the full RoPE matrix is blockwise $2 \times 2$ and thus very sparse. Explicitly, in view of Eq (2), one can simply take:

$$\begin{aligned} \boldsymbol{z}_i \to &\left[\cos(\theta_1), \cos(\theta_1), ..., \cos\big(\theta_{\frac{d}{2}}\big), \cos\big(\theta_{\frac{d}{2}}\big)\right] \odot \boldsymbol{z}_i \\ &+ \left[-\sin(\theta_1), \sin(\theta_1), ..., -\sin\big(\theta_{\frac{d}{2}}\big), \sin\big(\theta_{\frac{d}{2}}\big)\right] \odot \mathbf{P}\boldsymbol{z}_i. \end{aligned} \quad (5)$$

---

[1]For legibility, we generally suppress the dependence of $\text{RoPE}(\boldsymbol{r}_i)$ on $\{\boldsymbol{\omega}_n\}_{n=1}^{d/2}$, leaving it implicit.

[2]Given this property, some researchers taxonomise RoPE as a type of relative position encoding (RPE). However, we prefer to distinguish it as a separate class of PE, since PEs based on other high-dimensional rotations in $\text{SO}(d)$ are not necessarily translationally invariant (Schenck et al., 2025).

Here, $\odot$ denotes the Hadamard (element-wise) product. $\mathbf{P} \coloneqq \left[\delta_{\lfloor i/2 \rfloor, \lfloor j/2 \rfloor} - \delta_{i,j}\right]_{i,j=0}^{d-1} \in \{0,1\}^{d \times d}$ is the permutation that takes $\mathbf{P}\boldsymbol{x} = [\boldsymbol{x}_1, \boldsymbol{x}_0, \boldsymbol{x}_3, \boldsymbol{x}_2, ..., \boldsymbol{x}_{d-1}, \boldsymbol{x}_{d-2}]$, swapping alternate vector entries. Eq. (5) only needs $\mathcal{O}(d)$ operations. Moreover, it does not require the $N \times N$ attention matrix to be instantiated in memory. This is in contrast to regular bias-based RPE methods, which are generally $\mathcal{O}(N^2)$ and must instantiate attention in order to take $\boldsymbol{q}_i^\top \boldsymbol{k}_k \to \boldsymbol{q}_i^\top \boldsymbol{k}_j + b_{ij} \ \forall \ (i,j) \in \mathcal{N}^2$.[3]

**Expressivity of WIRE**. WIRE can distinguish graphs identical under the 1-dimensional Weisfeiler-Lehman graph isomorphism heuristic (with colours replaced by node features), because their adjacency matrices and hence node spectral coordinates differ. In this sense, transformers equipped with WIRE are *more expressive than standard GNNs*, which notoriously fail this test (Morris et al., 2019; Xu et al., 2018).

**Number of parameters**. The only learnable parameters in WIRE are the frequencies $(\boldsymbol{\omega}_i)_{i=1}^{d/2} \subset \mathbb{R}^m$, i.e. $dm/2$ parameters per transformer layer. Typically $m \ll d$, so this is very small compared to the rest of the network. For additional savings, one can share WIRE weights between layers or heads, or even follow conventional RoPE by freezing frequencies in an exponential decay pattern (Su et al., 2024).

**Generalising WIRE**. In this paper, we focus on instantiations of WIRE using spectral features. This is found to be effective in experiments (Section 4) and admits interesting theoretical analysis (Section 3.1) – in particular, recovering regular RoPE on grid graphs (Theorem 1) and exhibiting asymptotic dependence on graph effective resistance (Theorem 2). However, we emphasise that WIRE-like graph position encodings can in principle be implemented using *any* set of node features that capture structural information about $\mathcal{G}$, based on graph spectra, random walks or otherwise. This is important because the best position encoding may depend on the task and dataset at hand. Provided these features can be calculated in $\mathcal{O}(N)$ time, we also preserve compatability with linear attention.

**WIRE and GNNs**. In practice, for many graph-based tasks a combination of global attention and message passing layers gives the best performance, rather than a pure transformer (Rampášek et al., 2022; Shirzad et al., 2023). Naturally, WIRE is compatible with such hybrid models; one simply incorporates it wherever attention is used.

### 3.1 Properties of WIRE

WIRE enjoys a number of attractive theoretical properties. To start, note the following.

**Remark 1. (Equivariance under node ordering permutation)**. WIRE is insensitive to the choice of ordering of the nodes of the graph, up to sign flips and rotations of degenerate subspaces..

*Justification*. The spectrum $\{\boldsymbol{u}_k\}_{k=0}^{N-1}$ depends on the actual underlying graph structure $\mathcal{G}$; its entries are equivariant under permutation of the node ordering. The same follows for the WIRE transformation. Please see Section C.5 for important rebuttal clarifications.

**Theorem 1. (RoPE is a type of WIRE)**. RoPE is a special case of WIRE, occurring when one considers a grid graph $\mathcal{G}$ with specific learnable frequencies $\{\boldsymbol{\omega}_n\}_{n=1}^{\frac{d}{2}}$.

*Proof*. First consider a 1D grid (formally denoted as the path $P_N$), with adjacency matrix $\mathbf{A}_{ij} = \delta_{i,j+1} + \delta_{i,j-1}$. For this specific graph, the second (first nontrivial) eigenvector of $\mathbf{L}$ is given by $\boldsymbol{u}_1 = \left[-\cos\left(\frac{1}{N}\left(i + \frac{1}{2}\right)\pi\right)\right]_{i=0}^{N-1}$. This changes monotonically between $-\cos\left(\frac{\pi}{2N}\right)$ at $i = 0$ and $\cos\left(\frac{\pi}{2N}\right)$ at $i = N-1$. This sequence of coordinates, increasing as one progresses along $P_N$, is completely analogous to the token position coordinates $[0, 1, ..., N-1]$. They only differ by rescaling by $\frac{\pi}{N}$, offsetting by a constant, and restricting to the range $(-1, 1)$ by pushing through a cosine transformation. Taking $\boldsymbol{\omega}_i = [0, \omega_i, 0, 0, ...0]$, we isolate the

---

[3]Of course, exactly diagonalising $\mathbf{L}$ is generally $\mathcal{O}(N^3)$. Plenty of efficient approximate alternatives exist, e.g. the Lanczos algorithm (Lanczos, 1950). We describe our own novel variant in Section A.2. This standard one-time precomputation cost is incurred by any spectral PE method. For our purposes, the important takeaway is that WIRE can be used *without* instantiating $N \times N$ attention.

contribution from this first nontrivial spectral coordinate and recover regular RoPE used in LLMs, up to these simple bijective coordinate system transformations. See Figure 2 left.

Next, consider a 2 dimensional grid graph of size $N_x \times N_y$. This can be expressed as the Cartesian product graph $P_{N_x} \times P_{N_y}$, so the spectrum factorises. Completely analogously to the 1D case, the second and third eigenvectors are $\boldsymbol{u}_1[i] = \left[ -\cos\left( \frac{1}{N_x}\left(i_x + \frac{1}{2}\right)\pi \right) \right]_{i=0}^{N-1}$ and $\boldsymbol{u}_2[i] = \left[ -\cos\left( \frac{1}{N_y}\left(i_y + \frac{1}{2}\right)\pi \right) \right]_{i=0}^{N-1}$, with $i_y = \left\lfloor \frac{i}{N_x} \right\rfloor$ and $i_x = i - N_x i_y$. The order of $\boldsymbol{u}_1$ and $\boldsymbol{u}_2$ will depend on whether $N_x$ or $N_y$ is greater, but this detail is not important for our purposes. Taking $\boldsymbol{\omega}_i = \left[0, \omega_x, \omega_y, 0, ...\right]$, we now recover regular RoPE for ViTs. This is equivalent to applying 1D RoPE for each axis independently. See Figure 2 centre and right.

These arguments generalise straightforwardly to higher-dimensional grids (e.g. 3D for video), where one considers products of a progressively greater number of path graphs. ∎

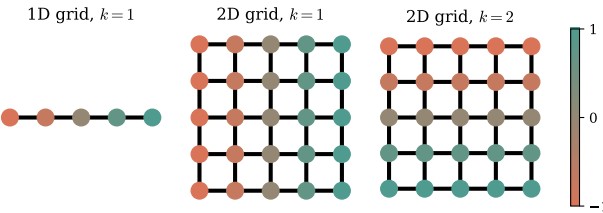

Figure 2: **RoPE $\subset$ WIRE.** The leading elements of the Laplacian eigenvectors of grid graphs (formally, Cartesian products of paths $P_N$) change monotonically in each direction. If we apply WIRE using just these coordinates, we recover regular RoPE as used in LLMs and ViTs. In this sense, RoPE is a special case of WIRE.

*Further comments on Theorem 1*. There are two minor differences between WIRE on (products of) path graphs and regular RoPE. First, as noted above, the spectral coordinates are always normalised to the range $(-1, 1)$, rather than taking values $0, ..., N-1$. This type of coordinate renormalisation is actually a popular trick in LLMs to improve generalisation with respect to sequence length (Chen et al., 2023; Li et al., 2023). It is intriguing that WIRE incorporates this regularisation automatically; we posit that it might improve generalisation to different graph sizes. Second, since the eigenvectors of $P_N$ are only unique up to a sign, one could equally flip the direction of all the spectral coordinates. This is not a property exhibited by RoPE when used in LLMs – here, there is a clear sense of directionality. Parity invariance follows from the fact that we consider undirected $\mathcal{G}$, so it is to be expected.

**Invariances under WIRE.** The commutativity and orthogonality of 2D rotations make RoPE translationally invariant: that is, $\left(\text{RoPE}(\boldsymbol{r}_i)\boldsymbol{q}_i\right)^\top \text{RoPE}(\boldsymbol{r}_j)\boldsymbol{k}_j = \left(\text{RoPE}(\boldsymbol{r}_i + \boldsymbol{c})\boldsymbol{q}_i\right)^\top \text{RoPE}(\boldsymbol{r}_j + \boldsymbol{c})\boldsymbol{k}_j \ \forall \ \boldsymbol{c} \in \mathbb{R}^m$. To rephrase, the composite transformation $\text{RoPE}(\boldsymbol{r_j} - \boldsymbol{r_i})$ applied to a query-key pair (implicitly in the case of linear attention) only depends upon the tokens' *separation* $\boldsymbol{r}_j - \boldsymbol{r}_i$, rather than their absolute positions. This property is important in 3D robotics applications (Schenck et al., 2025). It has been suggested to help sequence length generalisation in LLMs (Peng et al., 2023; Su et al., 2024).

WIRE automatically inherits the property described above. However, the interpretation of translational invariance in *spectral* space is less clear. Invariance under shortest path distance – a popular choice for RPE schemes made e.g. in Graphormer (Ying et al., 2021) – might be more intuitive. A closely-related alternative to shortest path distance is the *effective resistance* (Ellens et al., 2011; Velingker et al., 2023; Zhang et al., 2023), defined by

$$R(i,j) := \mathbf{L}^\dagger_{ii} + \mathbf{L}^\dagger_{jj} - 2\mathbf{L}^\dagger_{ij} \tag{6}$$

for nodes $(i, j) \in \mathcal{N}^2$. Here $\mathbf{L}^\dagger$ is the Laplacian pseudoinverse, which removes any diverging component of the regular inverse in the zero eigenvalue direction. It is straightforward to

confirm that $R(i, j)$ is a metric on $\mathcal{N}^2$. It is also known that effective resistance provides a lower bound for shortest path distance, with equality achieved for trees (Spielman, 2010).

**Theorem 2. (WIRE depends on resistive distance).** Consider a connected graph with spectral features $\boldsymbol{r}_i = \left[\boldsymbol{u}_k[i]/\sqrt{\lambda_k}\right]_{k=1}^{N-1} \in \mathbb{R}^{N-1}$. Suppose that we randomly sample the WIRE frequencies $\boldsymbol{\omega}_j \sim \mathcal{N}(0, \omega \mathbf{I}_{N-1})$, with $i = 1, ..., \frac{d}{2}$ and $\omega \in \mathbb{R}$. Given a query-key pair $(\boldsymbol{q}_i, \boldsymbol{k}_j) \in \mathbb{R}^d \times \mathbb{R}^d$, we have that

$$\mathbb{E}\left[(\mathrm{RoPE}(\boldsymbol{r}_i)\boldsymbol{q}_i)^\top \mathrm{RoPE}(\boldsymbol{r}_j)\boldsymbol{k}_j\right] = \boldsymbol{q}_i^\top \boldsymbol{k}_j(1 - \omega^2 R(i, j)/2) + \mathcal{O}(\omega^4), \tag{7}$$

where $R(i, j)$ is the effective resistance between nodes $i, j \in \mathcal{N}$. That is, in expectation, the leading contribution of WIRE is to downweight query-key logits by a factor proportional to the effective resistance.

*Proof.* See App. A.1. ∎

*Comments on Theorem 2* . We stress that WIRE is not *exactly* invariant under effective resistance for a particular draw of $(\boldsymbol{\omega}_i)_{i=1}^{d/2}$ due to (1) the $\mathcal{O}(\omega^4)$ correction terms and (2) the requirement of the expectation $\mathbb{E}(\cdot)$. In practice, we do not sample and average over an ensemble of random WIRE transformations, but instead take one learnable instantiation. Nonetheless, Theorem 2 builds intuition for how WIRE modulates the attention between pairs of nodes: the further apart they are, the more attention tends to be downweighted. It is remarkable that WIRE achieves this property *without needing to instantiate the attention matrix in memory*. One can (approximately) modulate the attention matrix entry $\boldsymbol{q}_i^\top \boldsymbol{k}_j \rightsquigarrow \boldsymbol{q}_i^\top \boldsymbol{k}_j(1 - \omega^2 R(i, j)/2)$, but without explicitly computing all $N \times N$ scores $\left\{\boldsymbol{q}_i^\top \boldsymbol{k}_j\right\}_{i,j=1}^N$ or resistances $\{R(i, j)\}_{i,j=1}^N$. This is of substantial interest for Performers. This type of principled 'linear attention topological masking' has long been a goal in the efficient transformer research community (Chen et al., 2023; Choromanski et al., 2022; Reid et al., 2024).

> **Takeaways from Section 3.1.** When considering the special case of grid graphs (formally, Cartesian products of path graphs $P_N$), WIRE recovers regular RoPE as used in LLMs and ViTs. If we instantiate WIRE with random weights, then the expected limiting transformation can downweight query-key logits depending upon their effective resistance – a lower bound to shortest path distance. Remarkably, WIRE exhibits this behaviour *without* needing to explicitly instantiate the attention matrix in memory.

## 4 EXPERIMENTS

Here, we test WIRE on a range of graph-based tasks, training **> 200 transformer models** in total. It provides a strong topological inductive bias, which often boosts performance.

### 4.1 SYNTHETIC TASKS: MONOCHROMATIC SUBGRAPHS AND SHORTEST PATHS

**Synthetic task 1. (Monochromatic subgraphs).** We begin with a synthetic task, chosen to strongly depend upon the structural properties of $\mathcal{G}$. We generate $10,000$ train graphs and $1,000$ test graphs with $N = 25$ nodes, beginning with a $5 \times 5$ grid and then deleting a randomly selected subset of edges. Every node is assigned a colour. We train a transformer to predict the size, i.e. number of nodes, of the largest monochromatic connected subgraph(s).

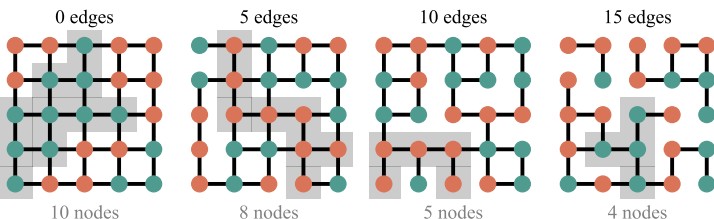

Figure 3: **Subgraph size regression.** Predict the number of nodes in the largest connected monochromatic subgraph(s) (shaded). Varying numbers of edges are removed, shown above.

**Choice of $\mathcal{G}$.** The motivation for constructing graphs as described above is that changing the number of deleted edges allows us to interpolate between 2D grid graphs and more complicated topologies. For grids WIRE can recover RoPE (Theorem 1), which is already known to perform well. The setup is similar to a ViT. On the other hand, as we delete more edges the topology becomes more complicated, testing how WIRE fares with trickier $\mathcal{G}$.

**Model details**. For the model inputs, we use the Laplacian eigenvectors, concatenated with node colour labels. This means that all our models include APE by default. For WIRE, we use spectral features using variable $m \in \{0, 3, 5, 10\}$. Clearly, $m = 0$ corresponds to *not* using WIRE. Growing $m$ incorporates progressively higher-frequency structural information into the rotations. Full architecture and training details are in Section B.1, along with visualisations of example attention patterns from the final transformer layer.

**Results.** Normalised test RMSEs are shown in Table 1, with standard errors in parentheses. WIRE provides gains over the baseline model ($m = 0$). When $\mathcal{G}$ is close to a grid (Figure 3 left), low-dimensional spectral features are sufficient. In contrast, as we delete more edges and $\mathcal{G}$ becomes more complicated (Figure 3 right), higher frequencies become helpful.

Table 1: **Monochromatic subgraph task**. Normalised test RMSEs for computing the largest monochromatic connected subgraph. $m$ is the spectral coordinate dimensionality; WIRE is used wherever $m > 0$. WIRE substantially improves regression performance.

| | Test RMSE ($\downarrow$) | | | |
| | Num. deleted edges | | | |
| $m$ | 0 | 5 | 10 | 15 |
| --- | --- | --- | --- | --- |
| 0 (no WIRE baseline) | 0.060(1) | 0.087(1) | 0.081(1) | 0.068(2) |
| 3 | **0.053(2)** | 0.075(2) | 0.072(3) | 0.064(3) |
| 5 | 0.057(2) | 0.075(1) | 0.070(2) | **0.056(4)** |
| 10 | 0.055(2) | **0.068(5)** | **0.063(2)** | 0.058(2) |

**Synthetic task 2. (Shortest path distances).** Next, we generate random Watts-Strogatz graphs with $N = 10$ nodes and $k = 2$ neighbours, with rewiring probability $p = 0.6$. Again, we take $10,000$ training examples and $1,000$ test examples. We train transformer models, identical to Task 1, to predict the shortest path distance (SPD) between two randomly selected nodes. Figure 4 *(left)* gives three examples, with the target and source nodes indicated in red and the corresponding SPD labelled above.

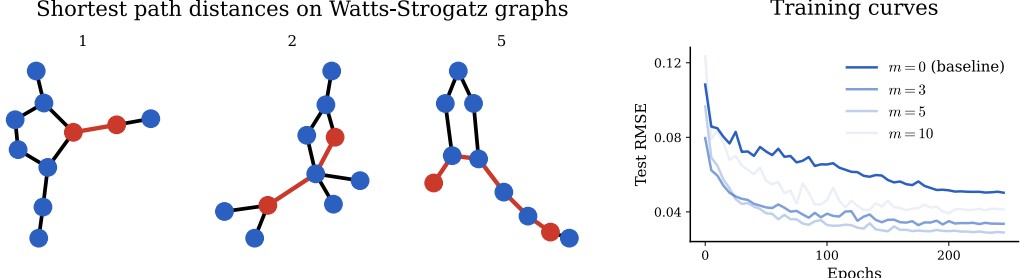

Figure 4: **Example Watts-Strogatz graphs for shortest path distance prediction.** *Left*: Random graphs labelled with shortest path distances between target and source nodes (red). *Right*: Corresponding training curves with $m \in \{0, 3, 5, 10\}$ spectral features.

Given WIRE's dependence on resistive distance (Theorem 2) – a lower bound to SPD – we expect it to provide a strong inductive bias. Table 2 confirms that this is indeed the case; WIRE nearly halves the test RMSE compared the APE-only baseline ($m = 0$). Figure 4 *(right)* shows sample training curves. App. B.1 gives further experimental details.

Table 2: **Shortest path distance task**. WIRE provides strong improvements to transformers trained to predict shortest path distances on random Watts-Strogatz graphs.

|  | Num. spectral coords, $m$ | | | |
|---|---|---|---|---|
|  | 0 (baseline) | 3 | 5 | 10 |
| **Test RMSE ($\downarrow$)** | 0.065(5) | 0.048(6) | **0.038(6)** | 0.045(4) |

**Number of parameters**. In all these models, the WIRE parameters constitute a tiny fraction of the entire model: **less than 1%** when $m = 3$. It is remarkable that they nonetheless lead to a strong performance boost. This spectral information is *already* being fed into the model as inputs. WIRE simply converts this into an additional strong structural inductive bias, applied throughout the network at every layer and every attention head.

### 4.2 Point cloud transformers

Next, we consider point cloud data (Guo et al., 2021). To implement WIRE, we construct a sparse $k$-nearest neighbours graphs. The input features are $(x, y, z)$ for each point. The following remark helps motivate graph-based position encodings in this setting.

**Remark 2. (Point cloud WIRE is invariant under SE(3) transformations)**. Trivially, the nearest neighbours graph $\mathcal{G}$ is invariant under joint translation and rotation of the point cloud data – namely, SE(3) transformations. The same follows for its spectrum, and thus the WIRE transformation we apply to queries and keys. Conversely, this property does *not* hold for RoPE transformations with 3D Cartesian coordinates, where rotation and translation will in general modify the position encoding.

**Classification and segmentation**. We train transformer models for classification and semantic segmentation, on the ModelNet40 (Sun et al., 2022) and ShapeNet (Chang et al., 2015) datasets respectively. Each example has 2048 points. We test (1) regular softmax attention, and (2) ReLU linear attention (a 'Performer') (Choromanski et al., 2020). Full details are in Section B.3. For WIRE, we use spectral features of dimensionality $m = 10$. The nearest neighbours graphs are constructed taking $k = 20$, which gives connected, sparse $\mathcal{G}$. As baselines, we include regular transformer and Performers without any additional position encoding ('NoPE'), as well as regular RoPE using Cartesian coordinates ('Cart. RoPE').

**Results**. The classification test metric is the precision of the object-level predictions (top one correctly classified). For semantic segmentation, it is the accuracy of the point-level predictions, weighted by the number of each each type of point. Table 3 gives the results. Runs are expensive, so following standard practice we report a single seed (Guo et al., 2021; Qi et al., 2017). WIRE outperforms the regular PCT (NoPE) baseline for both transformers and Performers, and often matches or surpasses Cartesian RoPE.

Table 3: **PCT results**. Test accuracy with different position encodings for classification and segmentation tasks, including both regular and efficient (Performer) attention. WIRE is consistently best (**boldface**) or second best (underlined), achieving greater accuracy than the regular PCT baseline (NoPE). It performs similarly to Cartesian RoPE, using $(x, y, z)$.

|  | Test accuracy ($\uparrow$) | | | |
|---|---|---|---|---|
|  | Classification (ModelNet40) | | Segmentation (ShapeNet) | |
| **PE** | Transformer | Performer | Transformer | Performer |
| NoPE | 91.8 | 90.1 | 93.1 | 92.8 |
| Cart. RoPE | 91.8 | **90.8** | **93.2** | **93.2** |
| WIRE | **93.4** | **90.8** | **93.2** | 93.0 |

### 4.3 WIRE Performers on benchmark tasks

Finally, we evaluate WIRE on established graph-based benchmarking tasks. To showcase its compatability with linear attention, we mostly focus on $\mathcal{O}(N)$ Performer models.

**WIRE is a drop-in addition to existing models**. For a clean, competitive implementation, we incorporate WIRE into GraphGPS architectures known to perform well on each benchmarking task (Rampášek et al., 2022). These are idiosyncratic; the best combination of message passing, attention and MLPs depends upon the particular task at hand. We use ReLU linear attention. Full details are in Section B.4. Remarkably, across the board, adding WIRE – a lightweight, extra structural inductive bias – can improve performance by multiple points. Whilst the linear variant still often performs worse than its expensive full-rank counterpart (the price of greater efficiency), we observe that WIRE is frequently able to *substantially close this gap*. For instance, on MalNet-Tiny, WIRE Performers are just as effective as transformers, but unlike the latter we can train on a single T4 12GB GPU.

Table 4: **Graph benchmark tasks**. Performer test metrics with and without WIRE, on graph benchmarks. (↑)/(↓) indicates whether higher or lower scores are better. For comparison, less efficient $\mathcal{O}(N^2)$ baselines from Rampášek et al. (2022) are also shown in gray.

| | Test metric | | |
|---|---|---|---|
| | Performer $\mathcal{O}(N)$ | | Transformer $\mathcal{O}(N^2)$ |
| **Dataset** | Baseline | WIRE | Baseline |
| MNIST (↑) | 97.56(2) | **98.10(1)** | 98.05(4) |
| CIFAR10 (↑) | 70.61(4) | **71.15(3)** | 72.3(1) |
| PATTERN (↑) | 85.71(3) | **86.63(6)** | 86.69(2) |
| CLUSTER (↑) | 76.90(3) | **77.53(3)** | 78.02(6) |
| ogbg-molhiv (↑) | 0.776(2) | **0.785(2)** | 0.788(1) |
| ogbg-molpcba (↑) | 0.238(3) | **0.264(1)** | 0.291(3) |
| ogbg-ppa (↑) | 0.8009(8) | **0.804(2)** | 0.802(3) |
| ogbg-code2 (↑) | 0.1731(9) | **0.1733(9)** | 0.189(2) |
| Peptides-func (↑) | 64.4(1) | **64.9(1)** | 65.4(4) |
| Peptides-struct (↓) | 0.2616(4) | **0.2566(4)** | 0.2500(5) |
| PascalVOC-SP (↑) | 0.367(1) | **0.376(1)** | 0.37(1) |
| MalNet-Tiny (↑) | 92.81(5) | **93.46(2)** | 93.36(6) |

**WIRE beyond Performers**. WIRE can be used within *any* model applying attention on $\mathcal{G}$. For example, WIRE often also provides gains when used with $\mathcal{O}(N^2)$ softmax attention, as noted in Section 4.1 and Section 4.2. We give further examples for a subset of the GNN benchmark datasets (smaller instances, where poor scalability is not prohibitive) in Table 8 of App. B.4. Equally, WIRE can be used within other efficient transformers like SGFormer (Wu et al., 2023) and BigBird (Zaheer et al., 2020) (the latter combined with GNNs within GPS), again improving test accuracy. See Table 9 in App. B.4. These short demonstrations provide further evidence of WIRE's broad utility. We defer exploration with yet more variants – such as Exphormers (Shirzad et al., 2023), which use virtual global nodes and expander graphs, and Graph Attention Networks (Veličković et al., 2017), which use local attention – as important future work.

> **Takeaways from Section 4.** WIRE provides a structural inductive bias the boosts the accuracy of transformers on graph-structured data. This includes in synthetic and point cloud settings, as well as more conventional GNN benchmarks.

## 5 CONCLUSION

We introduced Wavelet-Induced Rotary Encodings (WIRE), a new RoPE-style position encoding for graph-structured data. WIRE injects topological information into transformers by rotating tokens. Unlike many graph position encodings (e.g. Graphormer (Ying et al., 2021)), it is compatible with linear attention. In experiments, we find WIRE to be effective in tasks where a strong structural inductive bias is important.

**Reproducibility statement**. We have made every effort to ensure the work's reproducibility. The core algorithm is presented clearly in Alg. 1. Theoretical results are proved with accompanying assumptions in the main body and in App. A.1. Anonymised code is available here: https://anonymous.4open.science/r/WIRE_Graphs-4584/. It builds upon existing public repositories. The datasets in Section 4.2 and Section 4.3 are standard and freely available online. Exhaustive experimental details about the training and architectures are reported in App. B.

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

# A  Theory

This section contains extra proofs and comments considered too long for the main text.

## A.1  Proof of Theorem 2

Here, we show that randomly initialised WIRE tends to downweight attention scores, depending upon the resistive distance between the respective nodes.

*Proof.* For connected graphs, $\mathbf{L}^\dagger = \sum_{k=1}^{N-1} \frac{1}{\lambda_k} \boldsymbol{u}_k \boldsymbol{u}_k^\top$ since $\lambda_0 = 0$ but $\lambda_k \neq 0$ for $k \geq 1$. It is straightforward to see that $R(i,j) = \sum_{k=1}^{N-1} \frac{1}{\lambda_k} (\boldsymbol{u}_k[i] - \boldsymbol{u}_k[j])^2$. For each node $i \in \mathcal{N}$, we define an $N-1$-dimensional spectral feature $\boldsymbol{r}_i = \left[ \boldsymbol{u}_k[i]/\sqrt{\lambda_k} \right]_{k=1}^{N-1} \in \mathbb{R}^{N-1}$, whereupon $R(i,j) = \|\boldsymbol{r}_i - \boldsymbol{r}_j\|_2^2$. Considering random weights[4] $\boldsymbol{\omega}_i \sim \mathcal{N}(0, \omega \mathbf{I}_{N-1})$,

$$\mathbb{E}\left( (\boldsymbol{\omega}^\top \boldsymbol{r}_i - \boldsymbol{\omega}^\top \boldsymbol{r}_j)^2 \right) = \omega^2 \|\boldsymbol{r}_i - \boldsymbol{r}_j\|_2^2. \tag{8}$$

Given a query-key pair $(\boldsymbol{q}, \boldsymbol{k})$ at positions $(\boldsymbol{r}_i, \boldsymbol{r}_j)$,[5]

$$\boldsymbol{q}^\top \boldsymbol{k} \rightarrow \boldsymbol{q}^\top \text{RoPE}(\boldsymbol{r}_i)^\top \text{RoPE}(\boldsymbol{r}_j) \boldsymbol{k} =$$

$$\sum_{k=1}^{\frac{d}{2}} (\boldsymbol{q}_{2k-2} \boldsymbol{k}_{2k-2} + \boldsymbol{q}_{2k-1} \boldsymbol{k}_{2k-1}) \cos(\boldsymbol{\omega}_k^T (\boldsymbol{r}_i - \boldsymbol{r}_j)) \tag{9}$$
$$+ (\boldsymbol{q}_{2k-1} \boldsymbol{k}_{2k-2} - \boldsymbol{q}_{2k-2} \boldsymbol{k}_{2k-1}) \sin(\boldsymbol{\omega}_k^T (\boldsymbol{r}_i - \boldsymbol{r}_j)).$$

Taylor expanding in $\omega$ and taking the expectation,

$$\mathbb{E}(\boldsymbol{q}_i^\top \boldsymbol{k}_j) \rightarrow \boldsymbol{q}_i^\top \boldsymbol{k}_j \left( 1 - \frac{\omega^2}{2} \|\boldsymbol{r}_i - \boldsymbol{r}_j\|_2^2 \right) + \mathcal{O}(\omega^4) = \boldsymbol{q}_i^\top \boldsymbol{k}_j \left( 1 - \frac{\omega^2}{2} R(i,j) \right) + \mathcal{O}(\omega^4) \tag{10}$$

as claimed. Here, we used the fact that sin is an odd function to drop the $\mathcal{O}(\omega^3)$ terms. ∎

## A.2  Efficient diagonalisation of the Laplacian matrix via random features

In this appendix, we describe a new stochastic approximation algorithm for computing the leading eigenvalues and eigenvectors of the Laplacian matrix $\mathbf{L}$. This is a well-studied problem in the literature. We consider graphs $\mathcal{G}$ defined implicitly, where the edge weights are a function of the distance between nodes in some suitable metric space.

Recall that the (unnormalised) Laplacian is defined by

$$\mathbf{L} = \mathbf{D} - \mathbf{A}. \tag{11}$$

Suppose the adjacency matrix $\mathbf{A} = [a_{i,j}] \in \mathbb{R}^{N \times N}$ is defined by $a_{ij} = f_\theta(\|\boldsymbol{v}_i - \boldsymbol{v}_j\|_2)$, with $f_\theta$ is some (potentially learnable) function. The diagonal matrix $\mathbf{D}$ satisfies $d_{ii} = \sum_{j=0}^{N-1} a_{ij}$. Graph nodes are associated with coordinates in $\mathbb{R}^d$, e.g. $d=3$ for point clouds. For instance, for $\varepsilon$-ball graphs, one would take

$$f_\theta(\|\boldsymbol{v}_i - \boldsymbol{v}_j\|_2) = \mathbb{I}(\|\boldsymbol{v}_i - \boldsymbol{v}_j\|_2) \leq \varepsilon), \tag{12}$$

with $\mathbb{I}(\cdot)$ the indicator function. Denote $g(\boldsymbol{z}) = f(|\boldsymbol{z}|)$. We can rewrite $g$ as follows, for $i^2 = -1$:

$$g(\boldsymbol{z}) = \int_{\mathbb{R}^d} \exp(-2\pi i \omega^\top \boldsymbol{z}) \tau(\omega) d\omega, \tag{13}$$

---

[4]This is nothing other than the celebrated Johnson-Lindenstrauss transformation (Dasgupta and Gupta, 2003), a random projection that preserves vector norms and distances in expectation.

[5]In Eq. (9), we drop the $i$ and $j$ suffixes on the queries and keys for compactness, freeing it up to represent the coordinate $k \in \{0, ..., d-1\}$.

where $\tau$ is the inverse Fourier Transform of $g$ defined by:

$$\tau(\omega) = \int_{\mathbb{R}^d} \exp(2\pi i \boldsymbol{x}^\top \omega) g(\boldsymbol{x}) d\boldsymbol{x}. \tag{14}$$

Thus, taking: $\boldsymbol{z} = \boldsymbol{v}_i - \boldsymbol{v}_j$, we can rewrite:

$$a_{i,j} = C \cdot \mathbb{E}_{p(\omega)} \big[ \exp(-2\pi i \omega^\top \boldsymbol{v}_i) \exp(2\pi i \omega^\top \boldsymbol{v}_j) \big], \tag{15}$$

where $p(\omega)$ is the probability distribution with density proportional to $\tau(\omega)$ and $C = \int_{\mathbb{R}^d} \tau(\omega) d\omega$.[6] The ability to efficiently (potentially approximately) sample from $p(\omega)$ unlocks the following low-rank decomposition:

$$a_{i,j} \approx \Lambda^1(\boldsymbol{v}_i) \big( \Lambda^2(\boldsymbol{v}_j) \big)^\top, \tag{16}$$

where for $\omega_1, ..., \omega_r$ are sampled independently at random from $p(\omega)$, with $r \in \mathbb{N}$ the number of random features. In particular, we take

$$\Lambda^1(\boldsymbol{v}) = \sqrt{\frac{C}{r}} \big( \exp(-2\pi i \omega_1^\top \boldsymbol{v}), ..., \exp(-2\pi i \omega_r^\top \boldsymbol{v}) \big), \tag{17}$$

$$\Lambda^2(\boldsymbol{v}) = \sqrt{\frac{C}{r}} \big( \exp(2\pi i \omega_1^\top \boldsymbol{v}), ..., \exp(2\pi i \omega_r^\top \boldsymbol{v}) \big). \tag{18}$$

It follows that we can unbiasedly approximate $\mathbf{L}$ as:

$$\mathbf{L} \approx \mathbf{X}\mathbf{Y}^\top, \tag{19}$$

for matrices $\mathbf{X}, \mathbf{Y} \in \mathbb{R}^{N \times (N+r)}$ with rows $X(i)$ and $Y(i)$ given as follows:

$$X(i) = \eta_i \big( \sqrt{d_{ii}} \big) \oplus \Lambda^1(\boldsymbol{v}_i), \tag{20}$$

$$Y(i) = \eta_i \big( \sqrt{d_{ii}} \big) \oplus \big( -\Lambda^2(\boldsymbol{v}_i) \big). \tag{21}$$

Here, $\oplus$ denotes concatenation of the respective vectors, and $\eta_i(x)$ is a one-hot vector whose $i$th element is equal to $x$.

To reduce the dimensionality of the features, we can then apply standard Johnson-Lindenstrauss transformation (JLT). We unbiasedly approximate $\mathbf{X}\mathbf{Y}^\top$ by $\mathbf{X}'(\mathbf{Y}')^\top$, where the matrices $\mathbf{X}', \mathbf{Y}' \in \mathbb{R}^{N \times m}$ are given by:

$$\mathbf{X}' = \frac{1}{\sqrt{m}} \mathbf{X}\mathbf{G}, \quad \mathbf{Y}' = \frac{1}{\sqrt{m}} \mathbf{Y}\mathbf{G}. \tag{22}$$

The entries of the Gaussian matrix $\mathbf{G} \in \mathbb{R}^{(N+r) \times m}$ are drawn independently at random from the Gaussian distribution with mean $\mu = 0$ and standard deviation $\sigma = 1$.

We conclude that the Laplacian matrix $\mathbf{L}$ can be unbiasedly approximated as:

$$\mathbf{L} = \mathbf{X}'(\mathbf{Y}')^\top. \tag{23}$$

For $m \ll N$, this provides a computationally-efficient low-rank approximation.

Finally, applying results by Nakatsukasa (2019), we can efficiently compute the eigenvalues and eigenvectors of $\mathbf{X}'(\mathbf{Y}')^\top$ by diagonalising the smaller matrix $(\mathbf{Y}')^\top \mathbf{X}' \in \mathbb{R}^{m \times m}$. This operation only takes $\mathcal{O}(N)$ time, so it scales gracefully to very large graphs. It could be applied e.g. to the point cloud experiments described in Section 4.2, providing an alternative spectral approximation to the Lanczos algorithm. It may be of independent interest.

---

[6]We assume that this integral is well-defined.

# B    Extra experimental details

In this appendix, we provide extra experimental details to supplement Section 4.

## B.1    Synthetic experiments: monochromatic subgraphs and shortest paths

**Models and training**. For both tasks, we use a standard 4 layer transformer with model and MLP dimensionality 32. For simplicity, the attention is single-head. We train for 250 epochs with batch size 16, with a learning rate of $2 \times 10^{-4}$ obeying a cosine decay schedule ($\alpha = 0.01$). We train using the Adam optimiser with weight decay $1 \times 10^{-4}$. Dropout is applied at a rate of 0.2 to attention and the MLP outputs. Graph embeddings are obtained by mean pooling over node embeddings, and a dense layer projects the result to a scalar prediction for (1) the size of the largest monochromatic connected subgraph and (2) the shortest path distance between a target and source node (identified at the model inputs). Both datasets have $10,000$ training examples and $1,000$ test examples. We report the lowest test root mean squared error obtained during training, normalised by graph size. Standard errors are computed over 4 runs per setting.

**Ablation: WIRE attention patterns**. To better understand WIRE, we can also examine the activations of a trained model. For instance, Figure 5 shows rescaled attention scores at the final layer of the network. We take identical optimised weights, with WIRE either switched on as during training *(centre)* or off *(right)*. With WIRE, we see that nodes attend within the biggest monochromatic subgraph. The pattern disappears when WIRE is removed. This suggests that the network does indeed learn to use query-key rotations to carry structural information about $\mathcal{G}$.

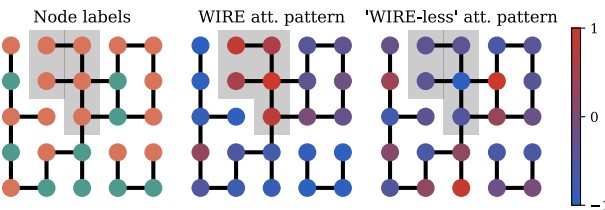

Figure 5: **Example attention patterns with WIRE.** Random choice of model input *(left)*, and example attention patterns for a trained model with *(centre)* and without *(right)* WIRE. WIRE helps nodes attend to other nearby nodes with the same label.

## B.2    WIRE and Performers

Recall that, for $\mathcal{O}(N)$ Performer attention, we take:

$$\boldsymbol{x}_i \to \frac{\sum_j \varphi(\boldsymbol{q}_i)^\top \varphi(\boldsymbol{k}_j) \boldsymbol{v}_j}{\sum_{j'} \varphi(\boldsymbol{q}_i)^\top \varphi(\boldsymbol{k}_{j'})}, \tag{24}$$

where $\boldsymbol{q}_i = \mathbf{W}_q \boldsymbol{x}_i$, $\boldsymbol{k}_i = \mathbf{W}_k \boldsymbol{x}_i$ and $\boldsymbol{v}_i = \mathbf{W}_v \boldsymbol{x}_i$ respectively, with $\mathbf{W}_q, \mathbf{W}_k, \mathbf{W}_v \in \mathbb{R}^{d \times d}$ learned projection matrices. $\varphi(\cdot) : \mathbb{R}^d \to \mathbb{R}^m$ is a (random) feature map, common choices for which include ReLU activations and random Laplace features (Yang et al., 2014).

There are two obvious manners in which one could incorporate WIRE:

1. *Directly modulating the queries and keys.* $\boldsymbol{z_i} \to \mathrm{RoPE}(\boldsymbol{r}_i)\boldsymbol{z}_i$ for $\boldsymbol{z}_i \in \{\boldsymbol{q}_i, \boldsymbol{k}_i\}$.

2. *Modulating the features.* $\varphi(\boldsymbol{z}_i) \to \mathrm{RoPE}(\boldsymbol{r}_i)\varphi(\boldsymbol{z}_i)$ for $\boldsymbol{z}_i \in \{\boldsymbol{q}_i, \boldsymbol{k}_i\}$.

The benefit of (1) is that, for suitable choices of maps $\varphi(\cdot)$ like ReLU, we have that

$$\varphi(\mathrm{RoPE}(\boldsymbol{r}_i)\boldsymbol{q}_i)^\top \varphi(\mathrm{RoPE}(\boldsymbol{r}_j)\boldsymbol{k}_j) \geq 0. \tag{25}$$

The attention scores all remain positive, which avoids instabilities caused by the denominator changing sign. Conversely, the advantage of (2) is that

$$(\text{RoPE}(\boldsymbol{r}_i)\varphi(\boldsymbol{q}_i))^\top (\text{RoPE}(\boldsymbol{r}_j)\varphi(\boldsymbol{k}_j)) = \varphi(\boldsymbol{q}_i)^\top \text{RoPE}(\boldsymbol{r}_j - \boldsymbol{r}_i)\varphi(\boldsymbol{k}_j), \qquad (26)$$

which gives the invariance properties we discuss in Section 3.1. But now modulated attention scores *can* be negative which can in general cause instabilities – something that Su et al. (2024) sidestep by only applying RoPE to the numerator (see Eq. 19 of their paper).

In Performer experiments, we find (1) to work well in practice, so tend to adopt this approach.

### B.3 Point cloud transformers

For classification, we consider the ModelNet40 dataset (Sun et al., 2022). Each includes 2048 points and and belongs to one of 40 object classes, including 'airplane', 'chair' and 'sofa'. The goal is to predict these labels. Meanwhile, for semantic segmentation we consider ShapeNet (Chang et al., 2015). Each point has an associated 'part label', breaking the object up into between 2 and 6 smaller semantically-meaningful sections – e.g. the legs or seat of a chair. The goal is to predict the class labels of each point.

**Models and training**. Building on the `Scenic` codebase (Dehghani et al., 2022),[7] we use a 4-layer transformer with hidden and MLP dimensions 128 and 512 respectively, trained for $10,000$ epochs with batch size 1024. We experiment with incorporating WIRE into only a subset of layers, anticipating that early layers that capture geometric information will benefit more from improved position encodings than the later semantic layers. This hyperparameter is optimised by a sweep. As baselines, we include regular transformer and Performers without any additional position encoding (NoPE), as well as regular RoPE using Cartesian coordinates (c.f. spectral). We train with the Adam optimiser, with weight decay 0.01. The learning rate schedule is compound (constant, cosine decay and linear warmup) with $10,000$ warmup steps and a base rate of $5 \times 10^{-6}$.

### B.4 GNN benchmark hyperparameters

In this section, we provide training details and hyperparameters for the GNN experiments reported in Section 4.3. We follow the setup of Rampášek et al. (2022). We choose MNIST, CIFAR-10, PATTERN and CLUSTER from 'benchmarking GNNs' (Dwivedi et al., 2020), Peptides-func, Peptides-struct and PascalVOC from the Long Range Graph Benchmark (Dwivedi et al., 2022), and ogbg-molhiv, ogbg-molpcba, ogbg-ppa and ogbg-code2 from the OGB datasets (Hu et al., 2020). We also consider MalNet-Tiny (Freitas et al., 2020). We provide the statistics for each dataset in Table 5.

Table 5: **Graph benchmark datasets**. Statistics of the datasets considered in Section 4.3.

| Dataset | # Graphs | Avg. nodes | Avg. edges | Dir. | Level / Task | Metric |
|---|---|---|---|---|---|---|
| MNIST | 70,000 | 70.6 | 564.5 | Yes | Graph, 10-class cls. | Accuracy |
| CIFAR10 | 60,000 | 117.6 | 941.1 | Yes | Graph, 10-class cls. | Accuracy |
| PATTERN | 14,000 | 118.9 | 3,039.3 | No | Inductive node, binary cls. | Accuracy |
| CLUSTER | 12,000 | 117.2 | 2,150.9 | No | Inductive node, 6-class cls. | Accuracy |
| ogbg-molhiv | 41,127 | 25.5 | 27.5 | No | Graph, binary cls. | AUROC |
| ogbg-molpcba | 437,929 | 26.0 | 28.1 | No | Graph, 128-task cls. | Avg. Precision |
| MalNet-Tiny | 5,000 | 1,410.3 | 2,859.9 | Yes | Graph, 5-class cls. | Accuracy |
| PascalVOC-SP | 11,355 | 479.4 | 2,710.5 | No | Inductive node, 21-class cls. | F1 score |
| Peptides-func | 15,535 | 150.9 | 307.3 | No | Graph, 10-task cls. | Avg. Precision |
| Peptides-struct | 15,535 | 150.9 | 307.3 | No | Graph, 11-task regression | MAE |

We follow the standard train/validation/test split in each case. For all datasets in 'benchmarking GNNs' and OGB – namely, MNIST, CIFAR-10, PATTERN, CLUSTER,

---

[7]See especially `https://github.com/google-research/scenic/tree/main/scenic/projects/pointcloud`.

ogbg-molhiv, ogbg-ppa and ogbg-molpcba – we run 10 seeds. Since MalNet-Tiny runs are expensive, we run 3 seeds. Likewise, the LRGB datasets – Peptides-func, Peptides-struct and PascalVOC-SP – are replicated 4 times. Lastly, all ogbg-code2 runs were repeated with 6 seeds. We use the AdamW optimiser (Loshchilov and Hutter, 2019) for all our experiments.

Our code is based on PyTorch Geometric. All experiments are run on a T4 GPU, with the exception of ogbg-ppa and ogbg-code2. The latter two datasets are much more compute intensive, and were run on an NVIDIA A100 (80GB) GPU. The results for the baseline dense transformer are taken from Rampášek et al. (2022), while the results for all other baselines are obtained from our own runs. The RoPE computation in Equation (25) is implemented using a learnable linear layer, transforming the spectral coordinates to dimensionality $d/2$. We control the scale of its initialisation with an additional hyperparameter.

### B.4.1 GraphGPS experiments: extra details

In this subsection, we provide further implementation details for all experiments using GraphGPS (Rampášek et al., 2022).

The ReLU-Performer model is described in Section B.2. For all our experiments, we default to the hyperparameters used by Rampášek et al. (2022). It is well established that performance is highly sensitive to the choice of hyperparameters for each dataset. For ogbg-ppa and ogbg-code2, all the hyperparameter settings were identical to (Rampášek et al., 2022, Table A.3), with optional 16 Laplacian positional encoding dimension for the WIRE Performer. We give details in Table 6.

Table 6: **GraphGPS Experiments with Performer Attention**. Hyperparameters used for our GraphGPS Experiments

| Hyperparame-ters | MNIST | CIFAR-10 | PATTERN | CLUSTER | Peptides-struct | Peptides-func | Pascal-Voc | MalNet-Tiny | ogbg-mol-hiv |
|---|---|---|---|---|---|---|---|---|---|
| Hidden Dim | 64 | 64 | 64 | 48 | 96 | 96 | 96 | 64 | 64 |
| Heads | 4 | 4 | 4 | 8 | 4 | 4 | 8 | 4 | 4 |
| Attention Dropout | .5 | .5 | .5 | .5 | .5 | .5 | .5 | .5 | .5 |
| MPNN | GINE | GatedGCN | GINE | GatedGCN | GatedGCN | GatedGCN | GatedGCN | GatedGCN | GINE |
| # Layers | 3 | 3 | 6 | 16 | 4 | 4 | 4 | 6 | 10 |
| GNN Dropout | .1 | 0. | 0. | .1 | .1 | .1 | .1 | 0. | 0. |
| Learning Rate | 0.0001 | .001 | 0.0005 | 0.0005 | .0003 | .0003 | .0005 | .001 | .0001 |
| Weight Decay | 1e-4 | 1e-5 | 1e-5 | 1e-5 | 0. | 1e-5 | 0. | 1e-5 | 1e-4 |
| # Laplacian Eigenvectors | 16 | 8 | 16 | 10 | 10 | 10 | 10 | 16 | 8 |
| # RWSE Fea-tures | 8 | - | 16 | - | - | - | - | 8 | 8 |
| Scheduler | ReduceLR | cos decay | cos decay | cos decay | cos decay | cos decay | cos decay | cos decay | ReduceLR |
| Batch Size | 64 | 64 | 32 | 32 | 128 | 128 | 32 | 4 | 32 |
| Laplacian Position Encod-ing Dim | - | 16 | - | 16 | 16 | 16 | 16 | - | - |
| Epochs | 150 | 150 | 100 | 100 | 200 | 150 | 300 | 150 | 100 |

Finally, following standard practice, for datasets like MNIST, PATTERN, MalNet-Tiny and ogbg-molhiv, we use random walks to provide global structural information. We use 16 walks for MalNet-Tiny and MNIST, and 20 walks for PATTERN and ogbg-molhiv. We also experiment with regular softmax and BigBird attention (Zaheer et al., 2020). In these cases, we again use the same hyperparameters. Details are provided below.

### B.4.2 SGFormer experimental details

SGFormer is another efficient transformer architecture, based upon a single linear attention layer and a single message passing layer (Wu et al., 2023). In contrast to our other Performer experiments, SGFormer takes the nonlinearity $\varphi(\cdot)$ to be the identity map. For message passing, we use a GCN. As usual, WIRE is injected into the attention mechanism of the transformer. Again, we mostly revert to the GraphGPS hyperparameters, avoiding extensive tuning to ensure our results are robust. Table 7 gives details.

Table 7: **SGFormer Experiments**. Hyperparameters used for the SGFormer Experiments.

| Hyperparameters | MNIST | CIFAR-10 | PATTERN |
|---|---|---|---|
| Hidden Dim | 128 | 256 | 128 |
| Heads | 2 | 1 | 8 |
| Attention Dropout | .5 | .5 | .5 |
| # GNN Layers | 3 | 2 | 3 |
| GNN Dropout | .1 | .1 | .1 |
| Learning Rate | 0.001 | .001 | 0.0005 |
| Weight Decay | 1e-5 | 0 | 1e-5 |
| # WIRE Features | 16 | 8 | 10 |
| Scheduler | ReduceLR | cosine decay | cosine decay |
| Epochs | 150 | 100 | 150 |
| Batch Size | 32 | 64 | 32 |

## B.5 EXTRA RESULTS FOR OTHER ATTENTION MECHANISMS ON GNN BENCHMARKS

Here, we report extra WIRE results with different (non-Performer) architectures, referenced in Section 4.3 of the main text. Specifically, we report results with regular softmax attention, SGFormer (Wu et al., 2023), and BigBird (Zaheer et al., 2020).

The SGFormer architecture is described above in Section B.4.2. Meanwhile, BigBird (Zaheer et al., 2020) combines local and global attention. It uses a small fixed number of global tokens that attend to all $N$ tokens. Remaining tokens attend to their neighbours. Table 8 and Table 9 shows that WIRE can be easily integrated these attention mechanisms, boosting the respective baselines.

Table 8: **WIRE results on softmax transformers**. Ablation results for WIRE on $\mathcal{O}(N^2)$ regular transformer architectures, on smaller datasets where poor scalability is not a problem. As observed in Section 4.1 and Section 4.2, our algorithm still improves performance.

| Dataset | Variant | Test metric | |
|---|---|---|---|
| | | Baseline | WIRE |
| MNIST ($\uparrow$) | Softmax transformer | 98.05(4) | **98.46(3)** |
| CIFAR-10 ($\uparrow$) | Softmax transformer | 72.3(1) | **73.48(7)** |
| PATTERN ($\uparrow$) | Softmax transformer | 86.69(2) | **86.75(2)** |
| CLUSTER ($\uparrow$) | Softmax transformer | 78.02(6) | **78.19(2)** |
| ogbg-molhiv ($\uparrow$) | Softmax transformer | 0.788(1) | **0.798(2)** |

Table 9: **WIRE results on extra efficient transformers**. Ablation results for WIRE on different $\mathcal{O}(N)$ transformer architectures: namely, SGFormer (Wu et al., 2023) and BigBird (Zaheer et al., 2020). Once more, WIRE can provide gains.

| Dataset | Variant | Test metric | |
|---|---|---|---|
| | | Baseline | WIRE |
| MNIST ($\uparrow$) | SGFormer | 96.78(4) | **97.3(1)** |
| CIFAR-10 ($\uparrow$) | SGFormer | 60.43(8) | **61.36(6)** |
| PATTERN ($\uparrow$) | SGFormer | 85.2(1) | **85.9(1)** |
| MNIST ($\uparrow$) | BigBird | 97.20 | **98.04** |
| CIFAR10 ($\uparrow$) | BigBird | 85.04 | **85.86** |

## C  ADDITIONS DURING REBUTTALS

### C.1  RWPE-WIRE

In the paragraph beginning 'generalising WIRE' (line 180), we noted that one need not necessarily use the Laplacian eigenvectors to compute the features $\{r_i\}_{i-1}^N \subset \mathbb{R}^m$ fed into RoPE. One could use other node features the capture the graph structure, such as *random walk position encodings* (RWPEs).

**RWPEs**. Considering an adjacency matrix $\mathbf{A}$ and a degree matrix $\mathbf{D}$, the random walk transition matrix is $\mathbf{P} := \mathbf{D}^{-1}\mathbf{A}$. The RWPE feature for node $i$ is

$$\text{RWPE}(v_i) := \left[\mathbf{P}_{ii}, \mathbf{P}_{ii}^2, \mathbf{P}_{ii}^3, ..., \mathbf{P}_{ii}^k\right] \in \mathbb{R}^k, \tag{27}$$

computing the probability of a random walk returning to node $v_i$ after $\{1, 2, ..., k\}$ steps. RWPEs are popular in the literature (Dwivedi et al., 2021; Rampášek et al., 2022). One can use RWPEs as rotational features for RoPE. Table 10 shows corresponding results (analagous to Table 2) for shortest path prediction, training with a single seed for 100 epochs. WIRE using graph spectra tends to perform better (and is in general more expensive), but we **also observe a gain over the no-WIRE baseline using RWPEs**. As in the main text, RWPEs are additionally provided as APEs, isolating the gains from RoPE rotations.

Table 10: **Shortest path distance task with RWPEs**. WIRE provides improvements to transformers trained to predict shortest path distances on random Watts-Strogatz graphs, using RWPEs instead of spectral features.

|  | | Num. spectral coords, $m$ | | |
| --- | --- | --- | --- | --- |
|  | 0 (baseline) | 3 | 5 | 10 |
| **Test RMSE ($\downarrow$)** | 0.061(1) | 0.060(1) | 0.059(1) | **0.055(2)** |

This demonstrates that WIRE is still an effective algorithm if graph spectra are not accessible. Investigating further features that are effective within WIRE is an interesting direction for future work.

### C.2  EXTRA GNN BENCHMARKS

We have added results for the large-scale graph benchmarks ogbg-ppa and ogbg-code2 to Table 4. Note that the gains for ogbg-code2 are very strong, with **Performer + WIRE achieving greater test accuracy than the softmax transformer baseline**.

### C.3  CLARIFICATION: DISTINGUISHING ISOSPECTRAL BUT NON-ISOMORPHIC GRAPHS

Isospectral but non-isomorphic graphs will have the same eigenvalues, but different eigenvectors. Since WIRE by default uses the eigenvectors (see Alg. 1), the WIRE transformation – and thus the transformer output – will be different. As such, WIRE can distinguish isospectral but non-isomorphic graphs.

### C.4  EFFICIENT DIAGONALISATION AND EXTRA DETAILS FOR SECTION A.2

**Time complexity of precomputation**. Below, we summarise the time complexity of common efficient diagonalisation algorithms used in the literature.

1. *Coarsening* (Loukas and Vandergheynst, 2018). These methods coarsen the graph (reduce $N$ to $N' \ll N$), compute eigenvectors on the small graph, and lift them back to the original graph. This is extremely fast for the lowest frequencies (smooth eigenvectors), and achieves good performance. It unlocks sub-linear time complexity relative to the original $N$ (after coarsening).

2. *Lanczos* (Baglama and Reichel, 2005; Lanczos, 1950). Once can compute the $m$ extreme eigenvalues using only matrix-vector multiplications. For sparse graphs, this is linear in the number of nodes $N$ for a fixed number of iterations. Modern improvements based on low-dimensional subspaces further improve efficiency.

3. *Other proxies*. More pragmatically, trading approximating eigenvectors with more general graph-based node features, one can compute other WIRE features in $\mathcal{O}(N)$ time using random walk position encodings (Section C.1) or using recent $\mathcal{O}(N)$ sparse methods like FastRP (Chen et al., 2019).

Halko et al. (2011) provides a detailed overview of other efficient randomised methods for computing low-rank decompositions of matrices like the graph Laplacian, also applicable to WIRE.

Lastly, we emphasise that some kind of structural feature is often *already computed* to be used as an absolute position embedding. In this case, one can also apply it via WIRE at essentially no extra cost.

**Time complexity of WIRE itself**. The time complexity of WIRE itself is $\mathcal{O}(Nmd)$ to project the features to dimensionality $d/2$ and $\mathcal{O}(Nd)$ to apply the sparse rotations. This is not observable in experiment wall-clock time, compared to the attention mechanism and MLPs. The memory footprint is tiny.

**Timing plots**. Figure 6 gives some example wall clock times for transformer forward passes with varying $m$, for the shortest path prediction task in Section 4.1. We use the same model hyperparameters as previously. Since the time complexity of projecting $m$-dimensional inputs to $d/2$-dimensional rotation angles for each token is $\mathcal{O}(Nmd)$, the plot is roughly linear in $m$ (deviating slightly due to hardware details and noise). We see that little cost is incurred by increasing $m$.

Note that we chose this toy example to show how the time complexity depends on $m$. In practical applications where $N$ and $d$ are much bigger (e.g. Section 4.3), the time incurred by applying RoPE rotations tends to be small compared to the attention and MLPs, as widely reported in the literature (Schenck et al., 2025; Su et al., 2024).

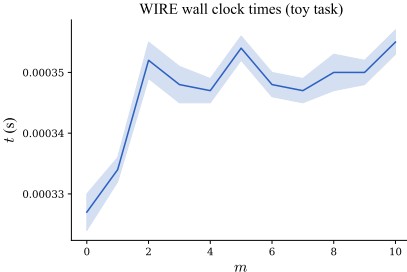

Figure 6: **Example attention patterns with WIRE.** Random choice of model input *(left)*, and example attention patterns for a trained model with *(centre)* and without *(right)* WIRE. WIRE helps nodes attend to other nearby nodes with the same label.

### C.5 Extra comments on invariance and equivariance

Note that, for the simplest instantiation of WIRE using the Laplacian eigenvectors, Remark 1 only holds up to sign flips and rotations of degenerate subspaces. Such transformations give vectors which are still eigenvectors of $\mathbf{L}$, but clearly the corresponding WIRE transformation can in general be different.

This is easily remedied by applying extra transformations to the spectral features to ensure that they are invariant under these transformations – for instance, maximal axis projection (Ma et al., 2023), sign flipping heuristics, or SignNet (Lim et al., 2022). In practice, we find that these additions make very little difference to our algorithm's empirical performance. We achieve our most competitive results (e.g. Table 4) using unmodified graph spectra.

**Intuition and asymptotic equivariance**. To understand this behaviour, we note that Theorem 2 *still holds* under random sign flips and basis transformations of the eigenvectors. Note that the leading term in Eq. (10) depends upon $\|\boldsymbol{r}_i - \boldsymbol{r}_j\|_2^2$, which is unmodified when these modifications are applied to $\{\boldsymbol{r}_i\}_{i=1}^N$. The fundamental asymptotic behaviour of (random) WIRE does not depend upon these ambiguities in basis and sign. It is intrinsically gauge invariant.

## C.6  WAVEPE-WIRE

To complement Section C.1, we can also use WavePE features (Khang Ngo et al., 2023) as rotational inputs to WIRE. These spectrum-based features use graph wavelets to capture multi-scale information.

**Constructing WavePE features.** Recall that we write the spectral decomposition of the Lapalcian as

$$\mathbf{L} = \mathbf{U}\boldsymbol{\Lambda}\mathbf{U}^\top, \quad \boldsymbol{\Lambda} = \mathrm{diag}(\lambda_0, ..., \lambda_{N-1}), \tag{28}$$

where $\mathbf{U}$ are the eigenvectors and $(\lambda_i)_{i=0}^{N-1}$ are the eigenvalues. We will consider a heat kernel filter function

$$g(s\lambda) = e^{-s\lambda}, \tag{29}$$

which is applied to the eigenvalues to create localised wavelents. For some scale $s \in \mathbb{R}$, the corresponding wavelet operator is

$$\varphi(s) = \mathbf{U}g(s\boldsymbol{\Lambda})\mathbf{U}^\top, \tag{30}$$

where $g$ is applied to each of the diagonal entries of the eigenvalue matrix. Concatenating a set of $k$ different scales $(s_i)_{i=0}^{k-1}$, we obtain the multi-scale diffusion tensor

$$\boldsymbol{\Psi} = [\varphi(s_i)]_{i=0}^{k-1} \in \mathbb{R}^{N \times N \times k}. \tag{31}$$

Further permutation-equivariant encodings are applied to map this to a set of $m$-dimensional features needed for WIRE. Many such transformations exist (Kondor et al., 2018; Maron et al., 2018), but in the interests of keeping the model lightweight we simply take:

$$\boldsymbol{r}_i = \mathrm{concat}\left(\boldsymbol{\Psi}[i, i, :], \sum_j \boldsymbol{\Psi}[i, j, :]\right) \in \mathbb{R}^{2k}, \; i \in \{1, ..., N\} \tag{32}$$

concatenating the diagonal entries of the tensor (self-diffusion) with its row sum (global-diffusion). As usual, these features are also linearly projected when passed to WIRE. It is straightforward to see that these features are **natively equivariant**, without any additional transformations.

**Empirical results**. One can directly replace WIRE's default spectral coordinates with the WavePE features defined in Eq. (32), e.g. for the shortest path prediction task. Trading our theoretical guarantees for these more empirical multi-scale features, we again see good performance in experiments; **like its regular counterpart, WIRE with WavePE consistently provides gains over the baseline**. Table 11 shows the results (companion to Table 2), ablating the dimension of the rotational features $m$. Note that, in this experiment, WavePE is only provided via WIRE, rather than as an APE. Given time constraints, we train for 100 epochs (c.f. Table 2).

Table 11: **Shortest path distance task with WavePE-WIRE**. Using WavePEs instead of raw eigenvectors as input features to WIRE also provides gains over the APE-only baseline.

| | | Num. spectral coords, $m$ | | |
|---|---|---|---|---|
| | 0 (baseline) | 3 | 5 | 10 |
| **Test RMSE ($\downarrow$)** | 0.080(1) | 0.077(2) | 0.073(1) | **0.071(1)** |

