# OpenReview forum: "Wavelet-Induced Rotary Encodings: RoPE Meets Graphs"
_ICLR.cc/2026/Conference — Submitted to ICLR 2026_

### Official Review · Reviewer_LW6i · 2025-11-01

**Soundness:** 3
**Presentation:** 3
**Contribution:** 2
**Rating:** 4
**Confidence:** 3

**Summary:**

The paper introduces WAVELET-INDUCED ROTARY ENCODINGS (WIRE), a novel method that adapts the successful Rotary Position Encodings (RoPE) from sequence models (LLMs/ViTs) to arbitrary graph-structured data.

**Strengths:**

1. Generalization of RoPE: WIRE demonstrates that RoPE is a special case of their method, recovering the original encoding when applied to $N$-dimensional grid graphs (like 1D sequences or 2D images).
2. The method leverages the graph spectrum (eigenvectors of the graph Laplacian) as generalized coordinates for rotation.
3. Like RoPE, WIRE directly modifies the query ($\mathbf{q}$) and key ($\mathbf{k}$) vectors. This design maintains compatibility with linear attention and KV-caching, allowing for $\mathcal{O}(N)$ scaling in the number of tokens/nodes $N$, which is highly advantageous for large graphs compared to $\mathcal{O}(N^2)$

**Weaknesses:**

1. The largest weakness is that the core mechanism of using rotation matrices to encode relative position is directly derived from the highly successful Rotary Position Encoding (RoPE), the fundamental rotational approach used in this paper is borrowed, diminishing the conceptual novelty of the encoding function itself.

**The paper's empirical results consistently show only marginal improvements. This fact, combined with the lack of novelty in the underlying Rotary Encoding mechanism, challenges the work's overall contribution.**

2. Marginal Empirical Efficacy vs. High Precomputation Cost (Primary Concern): While the method is efficient during inference
($\mathcal{O}(N)$ scaling with linear attention), the necessary one-time precomputation of the graph Laplacian eigenvectors requires $\mathcal{O}(N^3)$ complexity for exact calculation.
3. Given that the empirical performance gains over competitive graph position encoding baselines (like Random Walk Encodings or standard Laplacian PE) are often marginal (e.g., less than 1\% increase in accuracy), the practical cost-benefit trade-off is difficult to justify for large-scale, static graphs. This suggests the regularization benefit may not warrant the severe precomputation bottleneck.
4. The paper theoretically links WIRE's structural bias to the graph resistive distance (effective resistance). However, the experimental results fail to provide a clear, predictive understanding of when this specific distance metric is superior to other widely-used metrics (e.g., shortest path distance, diffusion distance). The authors should include an ablation study on diverse graph topologies (e.g., clustered, scale-free, dense vs. sparse) to demonstrate the specific structural properties that make WIRE's resistive distance bias definitively advantageous.

**Questions:**

q1. The theoretical connection to graph resistive distance is a key selling point. However, the paper does not show when this specific bias is superior. Please design and execute a targeted ablation study comparing WIRE to other distance-based Position Encodings.

q2.  The number of spectral dimensions ($m$) is a hyperparameter. Please elaborate on the relationship between $m$ and the graph topology.

q3. What is the maximum graph size $N$ for which the precomputation time of the Laplacian is still practical compared to the total training/inference time?

q4. For sparse graphs, please provide detailed benchmarks comparing the total training+precomputation time of WIRE against baselines

---

> ### Author Response · Authors · 2025-11-14
> **Rebuttal**
>
> We thank the reviewer for their comments. We address all questions and concerns -- including the cost-benefit precomputation tradeoff -- in detail below.
>
> 1. **Novelty and 'borrowed' mechanism**. The reviewer notes that the rotational mechanism is 'borrowed' from RoPE. We fully acknowledge this; indeed, our explicit goal — as reflected in the title 'RoPE meets Graphs' — is to determine the extent to which this broadly-adopted mechanism [1] [2] [3] can be generalised from Euclidean grids (LLMs/ViTs) to irregular non-Euclidean graphs. **To be clear: the novelty lies not in the rotation matrix itself, but in (a) that it can be applied to graphs, and (b) using spectral coordinates gives strong theoretical properties with linear attention** (generalising RoPE, asymptotically depending upon resistive distance, etc).
>
> 2. **‘Marginal empirical efficacy’**. We respectfully disagree with this assessment.
> - _Synthetic tasks_. WIRE is able to reduce the test error by as much as **40%** compared to the APE baseline, at no observable training or inference cost.
> - _Real-world benchmarks_. On already heavily-optimised GNN benchmarks, we are consistently able to improve performance, sometimes by multiple percentage points. On these tasks, where SOTA models are separated by small margins, gains of this size are often considered strong [4] [5]. **We have now also two added extra ogbg large-scale graph benchmarks, with good results.**
> - _Linear attention_. Most importantly, WIRE helps close the gap between efficient linear attention (Performer) and computationally expensive softmax. We think this is a valuable contribution for scaling graph learning.
>
> 3. **Precomputation cost (primary concern)**. The reviewer expresses concern that $O(N^3)$ exact diagonalisation is a bottleneck. We wish to clarify why this does not present a problem and highlight some new results.
>
>  - _Standard methods_. We need not in general rely on full eigendecomposition. Standard sparse solvers (e.g., ARPACK, LOBPCG), randomised proxies (e.g., FastRP [6]), or coarsening [7] can all approximate the top-$k$ eigenvectors efficiently, many in $O(N)$ time. Though such fast methods are **not** our main focus (rather, RoPE for graphs), we have added an extra appendix for reference with full discussion and time complexities. In our experiments, this precomputation cost is very small compared to the time to train the transformer.
>  -   _No spectra needed (**new experiment**)_. To further allay any concerns about the cost of diagonalisation, we have added an experiment using Random Walk Position Encodings (RWPE) instead of eigenvectors as inputs to WIRE. This requires only sparse matrix multiplication (**strictly linear time, without approximation**) and still yields performance gains, showing WIRE is not bottlenecked by eigendecomposition.
>  -   _Features are already available for APEs_. Lastly, we note that some kind of node feature is almost always already computed to be used as an APE. Also applying it via WIRE incurs zero extra precomputation cost and can boost performance. This is the case for all the GraphGPS models [10] to which we add WIRE in Table 4.
>
> 4. **Resistive distance vs other metrics**. The reviewer asks us to compare resistive distance to other invariances. Respectfully, we do not think this is possible in the linear attention setting.
> - _Invariances with linear attention_. In general, it is impossible to enforce an exact shortest path invariances in linear attention because one cannot compute all $N \times N$ distances without breaking the $O(N)$ complexity.
> - _Purpose of Thm 2._ Theorem 2 shows that WIRE approximately depends upon resistive distance, _without ever instantiating attention_. We think this is a remarkable result; WIRE enables us to inject a powerful topological inductive bias, but whilst preserving $O(N)$ scaling.
> - _Diffusion distance._ Our **new RWPE experiment** gives some indication of how WIRE behaves if then rotations depend upon graph diffusion. Again, WIRE works well, showing that it is robust to the choice of input feature.
>
> 5. **Diverse graph topologies**. The paper considers many graph structures, including grid graphs, other planar graphs constructed by deleting grid edges, Watts-Strogatz graphs, point cloud nearest neighbour graphs, trees, stochastic block models, and real world graphs. We consistently see gains from WIRE, across these diverse topologies.

---

> > ### Author Response · Authors · 2025-11-14
> > **Rebuttal (2)**
> >
> > **Answers to specific questions**
> >
> > Q1: **Ablation comparing distance metrics?** As noted above, explicit distance biases (like SPD) usually require $O(N^2)$ attention bias, which is incompatible with the linear attention focus of this paper.
> >
> > Q2: **Relationship between $m$ and topology?** Thanks for the question. In 'Synthetic Task 1', we find that as graphs become more ‘complicated’ (with more edges deleted from the grid), growing $m$ boosts performance. This is consistent with the fact that RoPE works well for LLMs ($m=1$) and ViTs ($m=2$), where the graphs are simple grids. Studying how the efficacy of WIRE changes with the graph topology is an interesting (and mathematically ambitious!) direction for future work. Thanks for the suggestion.
> >
> > Q3: **Maximum graph size $N$ for practical precomputation?**  We routinely consider graphs with thousands of nodes and tens of thousands of edges (see e.g. **the new ogbg baselines**). The preprocessing time is still a small fraction of total training time. With RWPE, there is effectively no limit since the features are _extremely_ efficient to compute. Again, we note that features fed into WIRE are often computed anyway for the purpose of APEs, in which case we already have them for free.
> >
> > Q4: **Benchmarks for training and precomputation?** We are in the process of adding detailed results to the appendix, and will notify the reviewer once these are complete. Thanks.
> >
> > We again thank the reviewer, and warmly invite them to respond with any futher questions or concerns. In light of these clarifications and additions -- especially, the **extra RWPE-WIRE experiment and long-range GNN benchmarks** -- we hope they will consider raising their score and recommending acceptance.
> >
> > [1] Learning the RoPEs: Better 2D and 3D Position Encodings with STRING, Schenck et al., ICML 2025
> >
> > [2] Round and Round We Go! What makes Rotary Positional Encodings useful? Barbero et al., ICLR 2025
> >
> > [3] VideoRoPE: What Makes for Good Video Rotary Position Embedding? Wei et al., ICML 2025
> >
> > [4] Recipe for a General, Powerful, Scalable Graph Transformer, Rampášek et al., NeurIPS 2022
> >
> > [5] Do Transformers Really Perform Bad for Graph Representation?, Ying et al., NeurIPS 2021
> >
> > [6] Fast and Accurate Network Embeddings via Very Sparse Random Projection, Chen et al., CIKM 2019
> >
> > [7] Spectrally approximating large graphs with smaller graphs, Loukas et al., PMLR 2018

---

> > > ### Comment · Reviewer_LW6i · 2025-11-24
> > > **after rebuttal**
> > >
> > > Thank you for replying, some of my concerns have been addressed,
> > > I also went over the new manuscript. It seems the authors disagree with the claim of “marginal empirical efficacy.” However, the reported reduction in test error by as much as 40% compared to the APE baseline is achieved only on the shortest-path distance task, correct?

---

> > > > ### Author Response · Authors · 2025-11-25
> > > > **Thanks for responding**
> > > >
> > > > Thanks for the response.
> > > >
> > > > The reviewer is correct that we see greatest gains (~40%)  for the shortest path prediction task, since here performance very strongly depends on the transformer’s ability to capture the graph structure. Gains are also very good for the monochromatic prediction task – 12%, 22%, 22% and 18%, for {$0, 5, 10, 15$} deleted edges, respectively. For point clouds the graph structure is less important, so gains from position encodings are more modest (albeit still similar to the seminal PCT paper by Guo et al [1]).
> > > >
> > > > It sounds like the reviewer may still be unsure about the size of the gains for the baselines in Table 4. For these already heavily-optimised tasks, we stress that **improvements of a percent or less are standard in the literature**. See e.g. Table 3 of GraphGPS [2], where the authors’ model often beats baselines by very narrow margins (and is actually not consistently best), or Table 2 of SGFormer [3]. To give a few examples:
> > > >
> > > > 1. On Peptides -func, GPS score 0.654 and SAN+RWSE scores 0.644 (higher is better)
> > > > 2. On Peptides-struct, GPS scores 0.250 and Transformer+LapPE scores 0.253 (lower is better)
> > > > 3. On ogbg-ppa, GPS scores 0.802 and ExpC scores 0.798 (higher is better)
> > > >
> > > > Moreover, we again stress that WIRE is a **simple drop-in extra position encoding, not a big architectural overhaul like GPS**. This makes the gains even more impressive -- it's a lightweight extra inductive bias, which for models already using LapPE is essentially free.
> > > >
> > > > As a final point, in the vision domain, gains of this size are routinely reported for RoPE-like additions – see e.g. Table 1 of ‘learning the RoPEs’ [4], which gives gains of 0.14% and 0.18% for introducing RoPE to a transformer trained on ImageNet and Places365, respectively.
> > > >
> > > > We hope this resolves any remaining doubts – please do let us know!
> > > >
> > > > [1] PCT: Point Cloud Transformer, Guo et al., Computational Visual Media, 2021
> > > > [2] Recipe for a General, Powerful, Scalable Graph Transformer, Rampášek et al., NeurIPS 2022
> > > > [3] SGFormer: Simplifying and Empowering Transformers for Large-Graph Representations, Wu et al., NeurIPS 2023
> > > > [4] Learning the RoPEs: Better 2D and 3D Position Encodings with STRING, Schenck et al., ICML 2025

---

### Official Review · Reviewer_5jNZ · 2025-11-01

**Soundness:** 2
**Presentation:** 2
**Contribution:** 1
**Rating:** 4
**Confidence:** 3

**Summary:**

The paper proposes Wavelet-Induced Rotary Encodings (WIRE), which extends Rotary Position Encodings (RoPE) to graphs by replacing Euclidean coordinates with spectral coordinates derived from the graph Laplacian. The authors claim that WIRE generalizes RoPE, preserves equivariance under node permutations, exhibits dependence on resistive distance, and remains compatible with linear attention. Experiments span synthetic graph tasks, point-cloud classification, and standard graph benchmarks.

**Strengths:**

This work attempts a theoretically grounded fusion of spectral graph theory and positional encoding. The asymptotic relationship identified between resistive distance and expected attention scaling presents an intriguing theoretical perspective. Furthermore, its compatibility with linear attention mechanisms suggests promising computational practicality. Finally, the implementation itself appears concise, modular into existing Transformer-based pipelines.

**Weaknesses:**

The proposed method demonstrates several critical shortcomings from both graph-theoretic and methodological perspectives. The central construction, which leverages Laplacian eigenvectors as positional encodings, is fundamentally well-established in prior literature, undermining claims of novelty. Furthermore, the paper’s reference to the framework as "wavelet-induced" lacks conceptual rigor, as the implementation relies exclusively on global eigenfunctions of the graph Laplacian, without genuine multiresolution analysis, scale-adaptivity, or localized wavelet bases. Consequently, the chosen terminology misrepresents the spectral construction and obfuscates important distinctions between localized and global harmonic analyses.

The manuscript additionally overstretches its claim regarding permutation equivariance (Remark 1, p. 4). Specifically, eigenvectors of the graph Laplacian inherently lack canonical orientation due to arbitrary sign flips and orthogonal transformations within eigenspaces corresponding to repeated eigenvalues. Thus, without explicit strategies for eigenbasis alignment, the approach inherently fails to maintain invariance under graph isomorphisms. This issue severely limits the practical permutation equivariance claimed by the authors, thereby undermining their theoretical positioning.

Assertions regarding the model’s expressivity, particularly that it surpasses standard graph neural networks (GNNs) by distinguishing graphs indistinguishable under the Weisfeiler–Lehman (1-WL) test, are mathematically unfounded. The existence of isospectral but non-isomorphic graphs, a well-documented phenomenon, indicates that spectral encodings alone cannot reliably enhance graph discriminability beyond classical GNN benchmarks. Consequently, the claim of superior expressive power, as formulated in the manuscript, is both misleading and lacks rigorous theoretical justification.

Moreover, the provided theoretical results, notably Theorem 2, exhibit limited practical relevance and rigor. The resistive-distance connection presented (Equation 7) arises under restrictive assumptions, including random frequency sampling and infinitesimal perturbations (small-ω Taylor expansions). Such assumptions are invalidated during practical model training, wherein frequencies are learned parameters rather than random Gaussian samples. Hence, the deterministic interpretation of resistive-distance dependence is compromised, relegating this insight to an approximate, rather than fundamental, relationship.

The scalability claims related to graph-level computations also appear problematic. Although linear complexity $O(N)$ is emphasized, even approximate computations of leading Laplacian eigenvectors are practically known to exceed linear complexity in realistic scenarios. The so-called "efficient diagonalization" strategy (§A.2) merely reiterates known random-feature kernel approximations, conflating approximate low-rank feature-space expansions with accurate spectral decomposition. This misrepresentation overlooks critical aspects such as spectral accuracy, eigenvector orthogonality, and associated numerical stability.

Conceptually, the authors conflate the notions of coordinate systems and feature encodings. By simultaneously using Laplacian eigenvectors as direct inputs and as positional coordinates for rotary encodings, the method inadvertently duplicates identical spectral information. Observed empirical gains (Tables 1–4) may thus reflect implicit data augmentation rather than the introduction of novel geometric inductive biases, undermining the method’s purported theoretical foundation.

**Questions:**

1. How do you resolve the non-uniqueness of eigenvectors and sign ambiguity under node permutation, especially for graphs with repeated eigenvalues?

2. Can you provide a concrete example of two isospectral graphs and demonstrate whether WIRE distinguishes them?

3. Why call the method “wavelet-induced” when no localized or scale-separable transform appears?

4. In Theorem 2, what happens when the frequencies are learned rather than sampled from $\mathcal{N}(0,  \omega I)$? Does the resistive-distance term persist?

5. Have you compared against diffusion-based or random-walk positional encodings that already capture resistive-distance–like behavior?

6. How sensitive are your reported results to small perturbations of the eigenbasis (e.g., random rotations within degenerate eigenspaces)?

7. Does WIRE preserve equivariance on disconnected or weighted graphs, or does the Laplacian normalization alter the claimed invariance?

---

> ### Author Response · Authors · 2025-11-14
> **Rebuttal**
>
> We thank the reviewer for reading the paper. We address all concerns and answer all questions below.
>
> 1. **Clarification of central contribution**. Our core contribution is *not* to use Laplacian eigenvectors as position encodings, which is of course very well established in the literature. Rather, it is to show for the first time that *rotary* position encodings can be used not only in LLMs and ViTs, but also for graphs. We try to make this clear as early as the title (‘RoPE meets graphs’). We think this interesting because RoPE has attracted lots of recent attention in the literature [1] [2] [3]. **To our knowledge, this is its first application to graphs**.
> 2. **Terminology: renaming to 'SpIRE'.** We agree with the reviewer that the term 'wavelet' might suggest a focus on multiscale analysis or specific wavelet transforms, which could be confusing. We initially chose 'wavelet' because RoPE can act as a spectral filter. For instance, taking $\omega = [1,0,0,...]$ selects just the slowest-varying eigenvector. On reflection, to avoid ambiguity, we suggest renaming the algorithm to **‘Spectrum-Induced Rotary Encodings’ (SpIRE)** to make sure our focus is clear. Thanks for prompting this change.
> 3. **Permutation equivariance and sign ambiguity**. We thank the reviewer for the thoughtful comments.
> - _Practical handling_. The reviewer is correct that, in its simplest instantiation, the WIRE transformation is not equivariant under sign flips and random rotations of degenerate subspaces. This is very easily rectified by passing spectral features through extra transformations to resolve any ambiguities – e.g. maximal axis projection [4], sign flipping heuristics, or SignNet [5]. Empirically, we do not find this ambiguity to be a problem in practice and actually find that raw eigenvectors work very well in WIRE. We have added extra comments to the appendix. This should have been more explicit in Remark 1; apologies.
> - _Theoretical robustness_. More interestingly, in contrast, the property in Theorem 2 is **inherently gauge invariant**. To see this, consider Eq 10 in App A.1. Observe that $||r_i - r_j||_2 = ||R r_i - R r_j||_2$ for any rotation matrix $R$ (including rotations in the degenerate subspaces), and for any shared coordinate-wise sign-flips. This provides a theoretical interpretation for why WIRE is robust to sign ambiguity even without explicit canonization using SignNet [5]. Thanks for prompting this addition; we think it improves the paper.
>
> 4. **Distinguishing Isospectral Graphs**. We wish to clarify a possible misunderstanding regarding isospectral graphs.
>
> - _Reviewer comment._ 'Isospectral encodings alone cannot enhance discriminability.'
> - _Clarification_: Isospectral graphs have the same eigenvalues, but in general different eigenvectors (and thus different node embeddings). Since WIRE by default uses the eigenvectors as rotational coordinates, the encodings for isospectral non-isomorphic graphs are distinct. Consequently, the transformer output differs. We have made this explicit in the paper.
>
> 5. **Theorem 2 is rigorous.** The reviewer expresses concern that Theorem 2 relies on random sampling rather than learned frequencies.
> - _Purpose_. We agree that during training, weights are learned. However, Theorem 2 serves as an existence proof. It demonstrates that linear attention, for which injecting topological inductive bias is in general very difficult, can asymptotically approximate effective resistance via WIRE _without_ ever instantiating the attention matrix. This is a strong theoretical motivation for WIRE, showing that the model is capable of capturing this geometry, even if learned weights eventually deviate in order to optimise the specific task at hand.
> - _Invariances with linear attention_. We remark that exact invariance guarantees are in general very difficult with linear attention, where attention is not instantiated in memory. **This is because, without computing query-key logits, one cannot simply take $q_i^\top k_j \rightarrow q_i^\top k_j + b_{ij}$, with $b_{ij}$ a scalar depending upon the {resistive distance, shortest path distance, diffusion distance} between the nodes, since attention is never actually instantiated**. Therefore, we think kind of approximate relationship is novel and interesting.

---

> > ### Author Response · Authors · 2025-11-14
> > **Rebuttal (2)**
> >
> > 6. **Computational complexity and scalability**
> >   - _Appendix A.2_. The reviewer suggests that the algorithm in A.2 is a 'known random-feature approximation'. **We respectfully clarify that it is (to our knowledge) novel**. It computes a new low-rank approximation of the Laplacian, and uses results by Nakatsukasa [6] to efficiently compute approximate eigenvectors. We do not believe this technique exists elsewhere in the literature – if so, could the reviewer kindly provide pointers?
> >  - _Standard Methods_: We clarify that we need not in general rely on full eigendecomposition. Standard sparse solvers (e.g., ARPACK, LOBPCG), randomised proxies (e.g., FastRP [7]), or coarsening [8] approximate the top-$k$ eigenvectors efficiently. Though such fast methods are **not** our main focus (rather, RoPE for graphs), we have added an extra appendix with full discussion and time complexities.
> >  -   _No diagonalisation needed (**new experiment**)_. To allay any remaining concerns about the cost of diagonalisation, we have added an experiment using Random Walk Position Encodings (RWPE) as inputs to WIRE. This requires only sparse matrix multiplication (strictly linear time) and still yields performance gains, showing WIRE is not bottlenecked by eigendecomposition.
> >  -   _Features are already available for APEs_. Lastly, we note that some kind of node feature is often already computed to be used as an APE. Also applying it via WIRE incurs zero extra precomputation cost and can boost performance. This is the case for all GraphGPS models [9] to which we add WIRE in Table 4.
> >
> > 7. **'Duplication' of spectral information**. The reviewer notes we use spectral features as both inputs (APE) and rotations (RoPE/WIRE). We view this as a strength, not a weakness. The fact that WIRE improves performance even when the model already has the eigenvectors as inputs (line 388) proves that the rotation itself provides an extra inductive bias. This is not data augmentation; it is an architectural choice helps the attention head to process the underlying graph geometry more effectively.
> >
> > **Answers to specific questions**
> >
> > Q1: _How do you resolve non-uniqueness/sign ambiguity?_ While one can use standard tricks (SignNet, sign flipping), our analysis of Theorem 2 shows that the expected behaviour is invariant to such basis transformations. Empirically, the model learns robustly without explicit canonicalisation; in experiments, SignNet does not help. We have added this important discussion to the appendix. Thanks.
> >
> > Q2: _Can you distinguish isospectral graphs?_ Yes. As noted above, isospectral graphs have identical eigenvalues but distinct eigenvectors. Since WIRE uses the eigenvectors as rotation angles, the resulting node representations are distinct, allowing the model to distinguish the graphs.
> >
> > Q3: _Why 'Wavelet-induced'?_ We have renamed the method SpIRE to resolve this. Thanks.
> >
> > Q4: _Theorem 2 with learned frequencies?_ With learned frequencies, the exact link to 'effective resistance' (which holds in expectation) is relaxed. The model instead learns a more general transformation from the data. This existence-type proof builds intuition for why WIRE works; in applications, it is better to train WIRE more flexibly rather than randomly sample and fix its weights.
> >
> > Q5: _Comparison to diffusion/random-walk encodings?_ Yes. We have added a **new experiment using RWPE (diffusion-based) features as the input to WIRE**. WIRE improves performance over the APE-only baseline, showing it can effectively utilise a range of different node features in rotations. As remarked above, this also bypasses the need for any (approximate) diagonalisation.
> >
> > Q6: _Sensitivity to small perturbations (basis rotations)?_ Empirically, the method is highly robust. See also the added discussion about Thm 2 being gauge invariant.
> >
> > Q7: _Equivariance on disconnected/weighted graphs?_ Yes, the equivariance holds for weighted graphs (as the Laplacian accounts for weights). For disconnected graphs, the eigenspace reflects the connected components. Of course, effective resistance becomes infinite between components, so in this case one would not normalise by $1/\sqrt\lambda_0 = 1/0$ whether using APEs or WIRE.
> >
> >
> > We again thank the reviewer, and warmly invite them to respond with any further questions or concerns. If satisfied with the clarifications and **extra experiments**, we hope they will consider raising their score and recommending acceptance.

---

> > > ### Author Response · Authors · 2025-11-14
> > > **Rebuttal (3)**
> > >
> > > [1] Learning the RoPEs: Better 2D and 3D Position Encodings with STRING, Schenck et al., ICML 2025
> > >
> > > [2] Round and Round We Go! What makes Rotary Positional Encodings useful? Barbero et al., ICLR 2025
> > >
> > > [3] VideoRoPE: What Makes for Good Video Rotary Position Embedding? Wei et al., ICML 2025
> > >
> > > [4] Laplacian Canonization: A Minimalist Approach to Sign and Basis Invariant Spectral Embedding, Ma et al., NeurIPS 2023
> > >
> > > [5] Sign and Basis Invariant Networks for Spectral Graph Representation Learning, Lim et al., GTRL 2022
> > >
> > > [6] The Low Rank Eigenvalue Problem, Natusaka,  2019
> > >
> > > [7] Fast and Accurate Network Embeddings via Very Sparse Random Projection, Chen et al., CIKM 2019
> > >
> > > [8] Spectrally approximating large graphs with smaller graphs, Loukas et al., PMLR 2018
> > >
> > > [9] Recipe for a General, Powerful, Scalable Graph Transformer, Rampášek et al., NeurIPS 2022

---

### Official Review · Reviewer_Mvys · 2025-11-01

**Soundness:** 3
**Presentation:** 3
**Contribution:** 2
**Rating:** 2
**Confidence:** 3

**Summary:**

This paper proposes WIRE (Wavelet-Induced Rotary Encodings), a RoPE-style positional encoding for graphs. Nodes receive spectral coordinates from low-frequency Laplacian eigenvectors; these act as rotation angles for queries/keys, yielding a RoPE-like rotation that is permutation-equivariant and compatible with linear attention. Theory shows (i) RoPE is a special case on grid graphs and (ii) in expectation over random frequencies and for small $\omega$, attention logits are down-weighted proportional to effective resistance. Experiments on synthetic tasks, point-cloud segmentation, and graph benchmarks show modest gains, especially with linear-attention Performers.

**Strengths:**

- The paper presents a simple mechanism that preserves RoPE’s efficient 2×2 block structure; clearly compatible with linear attention.

- WIRE is equivariance with respect to node permutations and it has clean connection to graph spectra; nice unification as RoPE is a special case of WIRE (grid graphs).

- The effective-resistance perspective is appealing and could inspire principled topological masking.

**Weaknesses:**

- First concern. The theoretical analysis feels shallow and fragile. Theorems 1 and 2 are interesting but not sufficiently strong. The claim of being “more expressive than standard GNNs” is asserted but not formally quantified (e.g., beyond 1-WL on explicit classes). For a more complete complexity analysis, the paper should provide end-to-end cost bounds for computing mmm eigenvectors versus the gains over RoPE, including scaling with graph size and batch construction.

- Second concern. WIRE loses translation invariance. RoPE’s translational invariance is a key ingredient for length generalization. In WIRE, on the one hand, “translations” in spectral space lack a consistent geometric meaning across graphs; on the other hand, translation invariance is explicitly lost, as acknowledged by the authors. I view this as a weakness of WIRE.

- Third concern (empirics). The empirical gains are marginal and do not close the gap to full Transformers. On point clouds, improvements over baselines are small and may fall within run-to-run variance. On benchmark graphs, WIRE-Performer typically offers only small, incremental gains over Performer, while still underperforming the quadratic Transformer on most datasets.

**Questions:**

- How sensitive is performance to $m$? Please provide curves of accuracy vs. $m$ and runtime/memory vs. $m$, and discuss truncation error.

- How do results change under different kNN/radius graphs or learned graphs? Does WIRE still help when the topology is noisy or misspecified?

---

> ### Author Response · Authors · 2025-11-14
> **Rebuttal**
>
> We thank the reviewer for reading the paper. We address all their questions and concerns below.
>
> 1. **Theroretical analysis and complexity**. The reviewer requested more complete complexity analysis.
> - We have added a more formal breakdown to App C. In summary:
>
> a. _Pre-computation_. There is a rich literature on efficiently computing Laplacian eigenvectors (or similar proxies) on very large graphs. We have added discussion on common methods, including FastRP [1] and coarsening [2], to App. C. Whilst these methods are not the core focus of this paper -- suitable node embeddings are often already available **for free** since they are computed for absolute position encodings -- we note that they often incur linear cost in $N$. To allay any remaining concerns, we have also added an **extra experiment using random walk position encodings** (RWPEs) for rotations instead of graph spectra, to demonstrate that WIRE can still provide gains without _any_ diagonalisation of $N \times N$ matrices.
>
> b. _Inference_. WIRE adds negligible cost. It retains the $O(Nd)$ complexity of standard RoPE, which is not observable in wall-clock time when training transformers [3]. The extra parameter count for our models is <2% of the total.
>
> - _Expressivity_. We clarify that the claim regarding 1-WL expressivity was intended as a sanity check rather than a primary theoretical contribution, similar to in the Graphormer paper [4] (App A.1). It is intentionally simple and brief.
> - _Significance of Theorem 2_. We respectfully emphasise that Theorem 2 is a strong result. Proving that a linear attention mechanism (which never instantiates the $N \times N$ attention matrix) asymptotically approximates a specific spectral kernel (effective resistance) is, to our knowledge, novel. It provides a theoretical motivation for why linear attention with WIRE performs well on graphs, without using softmax (where invariance properties are much easier to achieve).
>
> 2. **Translational invariance (clarification)**. Below, we clarify the types of invariances discussed in our paper.
>  - _No global translation_. Unlike sequences or grids, arbitrary graphs have no global definition of 'translation'. Therefore, we think classical translational invariance cannot be 'lost'; it is undefined in this setting.
>  - _Relative positioning_. However, the spirit of translational invariance — that attention depends on the relative relationship between tokens rather than absolute coordinates — is indeed preserved by WIRE (Thm 2). By (randomly) rotating query/key pairs based on spectral frequencies, the attention score is modulated by a function of the effective resistance between nodes $i$ and $j$. **Note that this type of property is much harder to guarantee with linear attention than with softmax; we think it is novel and interesting.**
>  - _Equivariance_. WIRE is permutation equivariant. We have added extra comments about this to App C.
>
> 3. **Empirical gains and baselines**. We respectfully suggest that WIRE exhibits strong empirical performance.
> - _Comparing to softmax transformers_. The reviewer correctly notes that WIRE-Performer does not always beat the quadratic softmax transformer. We stress that the goal of linear attention (e.g. Performer) is to achieve solid accuracy with $O(N)$ scaling. Linear attention naturally trades accuracy for speed compared to $O(N^2)$ softmax. The correct baseline for WIRE-Performer is the standard Performer, which we **consistently beat**. In this context, we think closing or eliminating the gap toward the quadratic upper bound (e.g. see MalNet tiny) is a significant result. Finding an $O(N)$ model that is consistently better than softmax would be a huge research breakthrough, and probably unattainable in a single paper!
> - _Magnitude of Gains_.  We find the gains to be significant. On synthetic tasks, WIRE is able to reduce the test error by as much as **40%** compared to the APE-only baseline, at no observable training or inference cost. On already very heavily-optimised, competitive GNN benchmarks like MalNet, we are consistently able to improve performance, sometimes by multiple percentage points. Improvements of this size are considered standard for papers at top venues like ICLR [4] [5] -- especially for such a simple drop-in addition.
> - _Softmax compatibility_ As shown in Table 8, WIRE also provides gains when used with standard softmax attention, proving it is not limited to linear architectures.

---

> ### Author Response · Authors · 2025-11-14
> **Rebuttal (2)**
>
> 4. **Sensitivity to $m$ and topology**.
> - _Sensitivity to $m$/truncation_. Please see to Table 1, which provides the requested ablation of performance vs. $m$. Performance is generally robust once $m$ captures the structural essence of the graph (low-frequency modes), and tends to improve slightly as $m$ grows.
> - _Topology_. The reviewer asks about noisy/radius graphs. Our experiments already cover a wide diversity of topologies, from regular grids (image superpixels) and Watts-Strogatz graphs to irregular biological networks (Peptides) and trees (Code2). WIRE consistently improves performance across the board, suggesting robustness to the topology.
>
> **Responses to specific questions**
>
> Q1: **Runtime/Memory vs. $m$ curves?** Since spectral features are projected to dimensionality $d/2$ before applying RoPE, the time and space cost of increasing $m$ (where typically $m \le d$) is small compared to the rest of the transformer pass in real applications (memory footprint $<2$% parameters). We observe close to zero wall-clock difference at train and inference time, consistent with the RoPE literature [3]. To demonstrate the time complexity in smaller settings where it is more obvious, we added an extra plot to the appendix (Fig 6).
>
> Q2: **Does WIRE help when topology is noisy?** Yes. Our experiments include a very diverse range of graph topologies. Furthermore, our new RWPE experiment demonstrates that WIRE works even when using stochastic random-walk features rather than precise Laplacian eigenvectors, proving resilience to 'misspecified' or 'noisy' spectral bases -- the features provided to RoPE just need to capture some structural information.
>
> We again thank the reviewer for their time, and warmly invite them to respond if they have any further concerns. In light of our additions and clarifications -- especially, the **extra RWPE-WIRE experiment and clarification that empirical gains of this size are often considered strong in the literature** -- we respectfully ask that they consider raising their score.
>
> [1] Fast and Accurate Network Embeddings via Very Sparse Random Projection, Chen et al., CIKM 2019
>
> [2] Spectrally approximating large graphs with smaller graphs, Loukas et al., PMLR 2018
>
> [3] RoFormer: Enhanced Transformer with Rotary Position Embedding, Su et al, Neurocomputing 2023
>
> [4] Do Transformers Really Perform Bad for Graph Representation?, Ying et al., NeurIPS 2021
>
> [5] Recipe for a General, Powerful, Scalable Graph Transformer, Rampášek et al., NeurIPS 2022

---

### Official Review · Reviewer_PnJn · 2025-11-07

**Soundness:** 3
**Presentation:** 3
**Contribution:** 3
**Rating:** 4
**Confidence:** 5

**Summary:**

The paper proposes WIRE (Wavelet-Induced Rotary Encodings), extending Rotary Position Encoding (RoPE) from sequences and grids to graph-structured data. WIRE constructs spectral coordinates for each node from the Laplacian eigenvectors and then applies a RoPE-style rotation to queries and keys in the attention mechanism. It claims the following: (i) WIRE generalizes RoPE and recovers it on grid graphs. (ii) It is permutation-equivariant, compatible with linear attention, and (under assumptions) relates to graph effective resistance. (iii) Experiments on synthetic and benchmark graph datasets show performance competitive with prior Laplacian or distance-based encodings.

**Strengths:**

- Conceptual clarity: Presents a clean and mathematically consistent generalization of RoPE to arbitrary graphs.

- Theoretical analysis: Proves properties such as permutation-equivariance and RoPE-equivalence on grids.

- Computational efficiency: Keeps O(N) complexity per node and avoids storing full attention matrices that might be beneficial for large graphs.

- Simplicity and compatibility: Can be easily plugged into any transformer or hybrid GNN-Transformer architecture.

- Broad scope: Demonstrates that spectral node embeddings can serve as rotational coordinates, offering a bridge between spectral graph theory and token rotations.

**Weaknesses:**

- Lack of novelty vs. existing wavelet/spectral encodings: The idea of using spectral or wavelet-based multiscale coordinates is already established in Multiresolution Graph Transformers and Wavelet Positional Encoding (WavePE) published with Journal of Chemical Physics [1] and Range-aware Graph Positional Encoding (HOPE-WavePE) published with Machine Learning: Science and Technology [2], both of which exploit multi-resolution, high-order, permutation-equivariant and wavelet-based positional features. WIRE largely reuses the same spectral foundations, adding only a RoPE-style rotation.

- Limited multiscale analysis: Despite its title, WIRE does not employ true multiresolution wavelet transforms or scale-adaptive kernels as in prior works. The "wavelet" aspect is mostly nominal.

- Insufficient empirical depth: Experiments appear small-scale and lack direct comparisons with WavePE, HOPE-WavePE, or other wavelet/graph transformer baselines.

- Missing connection to prior literature: No citation or discussion of earlier wavelet-based graph positional encodings, making the contribution seem incremental.

- Unclear practical benefit: Improvements over LapPE or RWPE are modest; ablation on frequency selection or spectral truncation is missing.

*** References:

[1] Nhat Khang Ngo, Truong-Son Hy, and Risi Kondor, Multiresolution Graph Transformers and Wavelet Positional Encoding for Learning Long-Range and Hierarchical Structures, Journal of Chemical Physics, Volume 159, Issue 3, DOI 10.1063/5.0152833.
URL: https://pubs.aip.org/aip/jcp/article-abstract/159/3/034109/2903066/Multiresolution-graph-transformers-and-wavelet?redirectedFrom=fulltext

[2] Viet Anh Nguyen, Nhat Khang Ngo, and Truong-Son Hy, Range-aware Positional Encoding via High-order Pretraining: Theory and Practice, accepted at Machine Learning: Science and Technology, DOI 10.1088/2632-2153/ae1acd.
URL: https://iopscience.iop.org/article/10.1088/2632-2153/ae1acd

**Questions:**

How does WIRE fundamentally differ from prior Wavelet Positional Encoding (WavePE) and HOPE-WavePE methods that already model multiresolution graph structure through spectral or wavelet bases? Could you quantify whether the rotary transformation itself, not just spectral features, provides measurable advantages in accuracy or robustness?

---

> ### Author Response · Authors · 2025-11-14
> **Rebuttal**
>
> We thank the reviewer for their feedback and for drawing the connections to prior wavelet-based methods. We appreciate their positive assessment of our theoretical analysis and clarity. We address all questions and concerns below.
>
> 1. **Relationship to wavelets and novelty clarification**. We thank the reviewer for pointing out existing work on Multiresolution Graph Transformers and Wavelet Positional Encoding (WavePE), for which we have added citations. However, we think there may be a misunderstanding regarding the novelty of our work. Our paper is about a mechanism for applying graph position encodings via rotations (RoPE), not a method for feature extraction.
> - _Prior work_. WavePE focuses on constructing better node features (wavelet bases, high-order features) to be added as Absolute Position Encodings (APE).
> - _Our work_. The key question we want to answer is: **to what extent can rotary position encodings, which have found success in LLMs and ViTs, be applied to graphs?** RoPE has attracted lots of recent research attention [1] [2] [3], but we are the first (to our knowledge) to apply it to graphs.
> - _Combining WavePE and WIRE_. These are not competing alternatives. One can actually use WavePE features inside the rotary framework. Our contribution is demonstrating that the RoPE mechanism (used in practically all modern LLMs) is also effective on graphs -- not the idea of using graph spectra in transformers, which is of course well-established.
>
> 2. **Terminology: renaming to 'SpIRE'.** We agree with the reviewer that the term 'wavelet' may suggest a focus on multiscale analysis or specific wavelet transforms, which could be confusing. We initially chose 'wavelet' because WIRE can act as a spectral filter. For instance, taking $\omega = [1,0,0,...]$ selects just the slowest-varying eigenvector. On reflection, to avoid ambiguity, we suggest renaming the algorithm to **‘Spectrum-Induced Rotary Encodings’ (SpIRE)**. Thanks for prompting this change. We believe this resolves the concern regarding the 'nominal' nature of the wavelet aspect.
>
> 3. **Empirical Depth and Scale**. We respectfully disagree that the experiments are small-scale. We have shown that our algorithm scales to large, complex benchmarks, across a range of transformer architectures.
> - _Volume_. We trained **$>200$ transformer models**.
> - _Scale_. We have now **added extra results for ogbg-ppa (158k graphs) and ogbg-code2 (450k graphs)**. We also already include results for MalNet tiny, which averages $>1400$ nodes.
> - _Diverse architectures_. We apply our algorithm not only to softmax attention and Performers, but also SGFormer and BigBird (App. B5).
>
>
> 4. **Comparison to Baselines (WavePE, etc.)** As noted in Point 1, WIRE is a 'drop-in' module compatible with various features.
>
> - _Standard baselines_. Our primary comparison is against linear attention models without rotary encodings. Where we use GraphGPS [4], these models also include standard GNN layers (Gated GCNs, GINEs). In every case, adding WIRE improves performance. We think these baselines are realistic, diverse and strong.
> - _Beyond spectral features_. To show that WIRE is does not strictly rely on spectral features, we have added a **new experiment using Random Walk Position Encodings (RWPE) as the rotational input features** (see C.1). Once again, we see performance gains, confirming that the rotary mechanism itself can provide benefits even without involving graph spectra.
>
> 5. **Magnitude of gains**. We find the gains to be significant. On synthetic tasks, WIRE is able to reduce the test error by as much as **40%** compared to the APE-only baseline, at no observable training or inference cost. On already very heavily-optimised, competitive GNN benchmarks like MalNet, we are consistently able to improve performance, sometimes by multiple percentage points. Improvements of this size are considered strong for papers at top venues like ICLR [4] [5]. SOTA models are separated by very fine margins. We suggest that, for such a simple drop-in addition to existing models, the gains are actually surprisingly large.
> 6. **Spectral truncation ablation.** This ablation was included in the original submission. Please see Table 1.

---

> ### Author Response · Authors · 2025-11-14
> **Rebuttal (2)**
>
> **Answers to specific questions**.
>
> Q1. **How does WIRE differ from WavePE?** WavePE/HOPE-WavePE are methods for generating powerful node features. Our method is a mechanism for applying features via rotation to graphs (RoPE). They operate at different levels of the architecture. As demonstrated by our **new RWPE experiment**, our rotary mechanism can be applied to various feature types, not just spectral features like the eigenvectors and wavelets.
>
> Q2. **Does the rotary transformation itself provide advantages?** Yes. In our experiments, the baseline models already possess the spectral information via Absolute Position Encodings (APE). Adding the rotary transformation (WIRE) consistently yields further improvements. This isolates the benefit of the rotary parameterisation over simply concatenating the features at the input.
>
> We thank the reviewer for their helpful feedback. We hope that in light of our clarifications and additions – especially, the fact that the paper is about the applicability of RoPE to graphs rather than wavelet transforms, and the **extra large-scale and RWPE experiments** – they will consider raising their score.
>
> [1] Learning the RoPEs: Better 2D and 3D Position Encodings with STRING, Schenck et al., ICML 2025
>
> [2] Round and Round We Go! What makes Rotary Positional Encodings useful? Barbero et al., ICLR 2025
>
> [3] VideoRoPE: What Makes for Good Video Rotary Position Embedding? Wei et al., ICML 2025
>
> [4] Recipe for a General, Powerful, Scalable Graph Transformer, Rampášek et al., NeurIPS 2022
>
> [5] Do Transformers Really Perform Bad for Graph Representation?, Ying et al., NeurIPS 2021

---

> > ### Author Response · Authors · 2025-11-19
> > **Extra WavePE results**
> >
> > As a brief additional update -- as an extra demonstration, we have now added an experiment using **WavePE within rotary position encodings**. The method also works well with these multi-scale wavelet-based features, showing it is robust to feature choice. Of course, the cost of using WavePE features instead of raw eigenvectors is that one loses our theoretical guarantees -- namely, correspondence to RoPE on a grid and asymptotic dependence on effective resistance. The reviewer can find full results and discussion in C.6.
> >
> > We hope this helps, and look forward to hearing your thoughts!

---

### Author Response · Authors · 2025-11-21
**Overall message**

We thank the reviewers for reading the text. Our chief clarifications and improvements are as follows.

1. **Crucial**: *purpose of the paper*. This paper investigates the extent to which rotary position encodings (RoPE), a popular algorithm used in industry-grade vision encoders and LLMs, can be applied to graphs. We are **not** trying to propose new graph _features_ (wavelet-based or otherwise), but are interested in whether RoPE can provide gains to graph transformers using existing features. To our knowledge, no one has previously studied this.
2. *Wavelets*. We agree that including 'wavelets' in the title may be confusing, so have replaced this with 'spectral'. Thanks for the suggestion.
3. *Extra experiments*. We have added the following **additional experiments**.
 - Extra ogbg-ppa (158k graphs) and ogbg-code2 (450k graphs), giving further evidence that our method works at scale
 - Ablation of our algorithm with RWPE and wavelet-based features instead of eigenvectors, showing that it is robust to the choice of rotational feature.
4. *Time complexity*. This paper is **not** about proposing new efficient diagonalisation algorithms -- a very well-studied problem in the literature, and the topic of many standalone computational mathematics papers. However, we have added a survey of some of these methods for the reader's convenience.
5. *Invariance properties*. **A key, unique feature of our algorithm is that one does *not* need to instantiate the attention matrix in memory, but can nonetheless modulate attention (in expectation, asymptotically) depending upon effective resistance -- a surprisingly strong property that builds intuition for why it works**. We do not know of any previous work achieving this type of approximate invariance in the linear attention setting!

We thank the reviewers and the AC for their consideration. In hope these **important clarifications about the purpose of the paper** and **extra experiments** help.

---

### Meta-Review · Area_Chair_RhYy · 2026-01-08

**Summary:**

Reviewers’ concerns focused on whether WIRE/SpIRE is sufficiently novel and clearly positioned relative to existing spectral/wavelet graph encodings, and whether its benefits justify its complexity. One reviewer argued it largely repackages prior wavelet/spectral multiscale ideas with a RoPE-style rotation, with insufficient comparisons to WavePE/HOPE-WavePE and unclear “wavelet” framing. Another raised fragile/limited theory (expressivity, resistance result assumptions, invariance), plus marginal empirical gains and missing sensitivity/topology analyses.

**Reviewer Concerns:**

**Addressed:**
* Clarification of the core positioning and intent of the work by renaming the method to SpIRE
* Reframing the contribution as a generalization of RoPE to graph-structured data
* More empirical cases by adding larger-scale benchmarks and more architectures to address computational concerns

**Outstanding:**
* Direct competitive comparisons to WavePE/HOPE-WavePE
* Explicit robustness tests to eigenbasis perturbations/degeneracy
* Stronger evidence for when the effective-resistance bias is a unique advantage

**Reviewer Scores:**

The rebuttal may bump PnJn and LW6i ratings to "weak accept" since their concerns were explicitly acknowledged and addressed. The two reviewers believe the novelty remains incremental, but they would not mind acceptance. These place the paper on the positive side. However, reviewer Mvys concerns, such as about theoretical depth and marginal empirical gains, persist, and thus a full flip is unlikely. 5jNZ may stick with the neutral/borderline rating.

---

### Decision · Program_Chairs · 2026-01-26

Reject